# Modulating the Blood–Brain Barrier: A Comprehensive Review

**DOI:** 10.3390/pharmaceutics13111980

**Published:** 2021-11-22

**Authors:** Rory Whelan, Grainne C. Hargaden, Andrew J. S. Knox

**Affiliations:** 1School of Biological and Health Sciences, Technological University Dublin, Central Quad, Grangegorman, D07 XT95 Dublin, Ireland; rory.whelan@tudublin.ie; 2Chemical and Structural Biology, Environmental Sustainability and Health Institute, Technological University Dublin, D07 H6K8 Dublin, Ireland; 3School of Chemical and Pharmaceutical Sciences, Technological University Dublin, Central Quad, Grangegorman, D07 XT95 Dublin, Ireland; grainne.hargaden@tudublin.ie

**Keywords:** blood-brain barrier (BBB), CNS drug delivery, tight junction (TJ), BBB permeability, BBB modulation, focused ultrasound, intra-arterial drug delivery, hyperosmolar agents, glioblastoma, alzheimers

## Abstract

The highly secure blood–brain barrier (BBB) restricts drug access to the brain, limiting the molecular toolkit for treating central nervous system (CNS) diseases to small, lipophilic drugs. Development of a safe and effective BBB modulator would revolutionise the treatment of CNS diseases and future drug development in the area. Naturally, the field has garnered a great deal of attention, leading to a vast and diverse range of BBB modulators. In this review, we summarise and compare the various classes of BBB modulators developed over the last five decades—their recent advancements, advantages and disadvantages, while providing some insight into their future as BBB modulators.

## 1. Introduction

The blood–brain barrier (BBB) is the term used to describe the fortified microvascular network of the CNS, noted for its remarkably heightened molecular specificity—facilitating nutrient supply, whilst impeding the access of harmful substances to the brain. BBB passage of endogenous biomolecules is generally achieved via transcellular and paracellular pathways. Transcellular processes facilitate the transport of molecules *through* the cell, utilising passive diffusion or a wide range of highly specific and diverse molecular transport systems involving transcytosis. Passive diffusion is a general route of access across the BBB, but certain restrictions prevent most molecules from successfully utilising this pathway. In the past 20 years, a range of lipid diffusion models have shown correlations with various physicochemical properties for predicting a drug’s BBB permeability. While certain models have been debated [1], a common trend amongst them is that a molecule must be lipophilic (LogP < 5) and small (<500 Da) to pass the BBB efficiently [2].

**Table 1 pharmaceutics-13-01980-t001:** Various Parameters for Predicting Small Molecule BBB permeability.

Physicochemical Properties	Value	Reference
Molecular weight	<500 Da	Lipinski et al. [2]
Molecular weight	<400 Da	Levin [3]
Molecular weight	<450 Da	Atkinson et al. [4]
LogP	<5	Lipinski et al. [2]
LogP	1.5–2.7	Hansch et leo [5]
LogD	1–3	Van de Waterbeemd et al. [6]
Hydrogen bond donors	<5	Lipinski et al. [2]
Hydrogen bond acceptors	<10	Lipinski et al. [2]
Hydrogen bonds	<8	Pajouhesh et Lenz [7]
Polar Hydrogen atoms	0–1	Ghose et al. [8]
No. Nitrogens	1–2	Ghose et al. [8]
No. Nitrogens + Oxygens	2–4	Ghose et al. [8]
Polar surface area (PSA)	<90 A^2^	Hitchcock et Pennington [9]
Polar surface area (PSA)	<60–70 A^2^	Kelder et al. [10]
Polar surface area (PSA)	25−60 Å^2^	Ghose et al. [8]
Solvent accessible surface area	455−575 Å^2^	Ghose et al. [8]
pKa	4–10	Fischer et al. [11]
Carboxylic acid functional groups	None, unless AA residue	Ghose et al. [8]
Rotatable bonds	<5	Pajouhesh et Lenz [7]
Rotatable bonds	1–4	Ghose et al. [8]
Molecular volume	740−970 Å^3^	Ghose et al. [8]
Cytochrome P450 Inhibition	<50% at 30 μM	Pajouhesh et Lenz [7]
CYP2D6 metabolism	Low	Pajouhesh et Lenz [7]
CYP3A4 inducer	Not potent	Pajouhesh et Lenz [7]
Serum albumin affinity	Kd < 10 μM	Raub et al. [12]
P-Glycoprotein affinity	None to low	Raub et al. [12]
Aqueous solubility	>60 μg/mL	Pajouhesh et Lenz [7]
Effective permeability	1 × 10^−6^ cm/s	Pajouhesh et Lenz [7]
CNS MPO	≥4	Wager et al. [13]

Abbreviations: AA (amino acid), CNS MPO (central nervous system multiparameter optimisation).

Considering the physicochemical properties governing BBB permeability presented in Table 1, the restrictions for passage across the BBB do not appear severely stringent. However, the presence of a defence system implemented by endothelial cells actively pumps out most xenobiotics and non-CNS essential endogenous biomolecules—known as efflux pumps. These integral membrane proteins primarily localise to the luminal side of endothelial cells and pump their substrates back into the bloodstream against a concentration gradient. From a mechanistic perspective, efflux pumps act like a reverse form of carrier-mediated proteins. Xenobiotics bind to efflux pumps within the cell, forming a protein–substrate complex. ATP causes the pump to undergo a conformational change which pushes the drug out of the cell and back into the blood. Although effective in preventing unwanted toxins from reaching the brain, it is also a major bottleneck in the transcellular shuttling of drugs across the BBB. Additionally, their presence is amplified drastically in brain endothelial cells relative to other endothelia [14].

Paracellular transport involves the passage of molecules *between* the gaps of neighbouring cells. This process is restricted and generally limited to diffusion of small ions and water molecules. The remarkably restrictive properties of the BBB’s paracellular flux is due to the high concentration of tight junctions (TJs) found within the intercellular cleft between neighbouring brain microvascular endothelial cells (BMECs). Tight junctions are composed of a branching network of ‘sealing strands’, with each strand formed between integral membrane proteins embedded in the membranes of neighbouring endothelial cells. The extracellular domains of each transcellular protein associate with one another, forming stable barriers that can also adapt to the environment and exhibit significant plasticity. Tight junction proteins include occludin, claudins and junctional adhesion molecules. The strength and rigidity of these proteins is due to their interactions on the inner side of the cell with a membrane-associated cytoplasmic protein; the most prominent of which are zonula occludens (ZOs). Zonula occludens act as scaffolding proteins by binding to both TJ proteins and the cytoskeleton via actin filaments. The resultant structural organisation creates a network of strongly embedded TJ proteins, giving rise to tightly sealed paracellular gaps (Figure 1).

Analysis of the Comprehensive Medicinal Chemistry Database revealed that only 5% of marketed drugs are inherently BBB permeable [18]. In fact, between 2010–2017, the probability of a CNS drug candidate succeeding in clinical trials was on average 12 times lower than non-CNS drugs [19] (Figure 2). While BBB permeability does not have a direct influence on a CNS drug candidates success in clinical trials [20], modulating the BBB would enable larger and more polar molecules to reach the CNS—enabling more effective drugs to enter clinical trials.

BBB regulation is a tightly controlled and rate-limiting factor in determining CNS penetration of pharmaceuticals for the treatment of diseases such as Parkinson’s disease, Alzheimer’s disease and glioblastoma. Consider doxorubicin; an FDA-approved antineoplastic chemotherapeutic used to slow extracranial tumour growth. Because of its affinity for multi-drug resistant efflux transporters [21] it fails to cross the BBB efficiently, rendering it an ineffective treatment for brain tumours. Doxorubicin’s effect on brain cancer however has been demonstrated by direct injection into brain tumours, with potency against glioma cells being reported as 2000 times greater than the current clinical standard Temozolomide treatment in vitro [22,23]. Other anticancer drugs such as Trastuzumab [24] do not cross the BBB at all, yet intracranial administration of this therapeutic has demonstrated alleviation of glioblastoma tumor progression.

BBB modulators have been under investigation for five decades [25] with multiple innovative approaches being developed over this time. These modulators either (1) exploit the cells vehicular machinery—hijacking *transcellular* uptake in a Trojan horse manner or (2) deconstruct cell–cell contacts, widening their tight junctions and facilitating *paracellular* transport of chemical species larger than small monoatomic ions. Both approaches have advantages and limitations. While transcellular modulators facilitate a more selective solute passage, drug uptake is far less efficient. Transcellular-mediated systems are dependent in both energy and active binding; thus, their kinetics are restricted. Conversely, paracellular modulation enhances passive broadscale passage of solutes, whereby it effectively ‘opens the floodgates’—the opening of which however can be attenuated. Here, we provide a comprehensive summary of the various BBB modulator classes, comparing their advantages and limitations while analysing their current status towards implementation in the clinic.

## 2. Current Agents for Modulating the BBB

### 2.1. Focused Ultrasound

Focused ultrasound (FUS) is an approach that utilises sound waves to induce a mechanical or thermal effect on targeted tissue without the need for radiation or surgical intervention [26]. Similar to light waves passing through a convex lens (i.e., a magnifying glass), ultrasound waves can be focused into a small area, where their intensity is amplified to a level that can interfere with biological tissue through either mechanical or thermal processes. The effect ultrasound has on its target is based on the power intensity of the sound wave applied. Low-intensity pulsed ultrasound (LIPUS) has applications in the stimulation of motor responses [27,28] and suppression of neural activtiy [29,30]. The underlying mechanism of action is thought to be “mechanical perturbation of voltage-dependant ion channels or changes in bilayer impedance” [31]. Slightly higher in intensity, focused ultrasound (FUS) induces cellular cavitation in endothelial cells by mechanical intervention, transiently opening the BBB and aiding drug delivery to the brain [32,33,34,35,36]. High-intensity focused ultrasound (HIFU) results in thermal, irreversible tissue destruction by inducing coagulative necrosis [31]. This technique has been applied to the treatment of essential tremor [37], Parkinson’s disease [38] and neuropathic pain [39].

### 2.2. Development of Focused Ultrasound BBB modulation

The disruption of endothelial cells by ultrasound was first reported in 1953 by Lehmann et Herrick [40] who reported the transient cavitation of blood vessels in mice; however, the disruption observed resulted in fatal brain haemorrhaging. Instrumental development and further research resulted in attenuation of these deleterious effects, whereby tissues could be selectively disrupted by adjustment of pulse intensity, frequency and duration [41]. Vykhodtseva et al.’s investigations demonstrated selective blood–brain barrier disruption (BBBD) without parenchymal damage being observed in vivo [41]. Further to this, low-frequency ultrasound pulses were shown to penetrate human skulls and create focal points of millimetric accuracy [42]. At the turn of the millennium, further improvements in FUS precision were observed by the incorporation of magnetic resonance imaging (MRI)-guided modulation. The combination of MRI and FUS enabled the target site to be visualised in real time, allowing a greater accuracy and precision in BBB disruption, and granting intraoperative target guidance. Additionally, magnetic resonance-guided FUS (MRgFUS) can confirm target site BBB modulation and uptake of a drug, if co-administered with a contrast agent [43]. Hynynen et al. also implemented microbubble-mediated FUS in this study. The microbubbles, upon ultrasound stimulation, rapidly expand and contract, disrupting the BBB and creating cavitation sites at a sensitivity greater than direct FUS (Figure 3). Importantly, this reduced the power required to induce transient BBBD by 100-fold; falling far below the levels required for tissue damage [43]. The effect of altering BBB permeability using microbubbles was reported previously by Hills et James in a study investigating the mechanism of action for decompression sickness [44]. In this study, microbubbles were injected into the right carotid artery of guinea pigs, which resulted in BBB disruption and focal edema in the ipsilateral region of the brain, as shown by paracellular traversal of trypan blue into the brain parencyhma [44]. Hynynen et al. however applied microbubble-assisted BBBD therapeutically—developing and launching MRgFUS as a viable candidate for BBB modulation [43].

#### 2.2.1. MRgFUS BBB Modulation—Targeted Drug Delivery and Gene Therapy

As previously mentioned, doxorubicin cannot cross the BBB and is an ineffective chemotherapeutic agent for glioblastoma, unless highly invasive procedures are employed [22]. The incorporation of liposomal-encased doxorubicin used in conjunction with MRgFUS however slowed tumour growth, with tumour doubling time increasing from 2.3 days to 3.7 days. Furthermore, an increased median survival time of 24% was observed in treated rats. Encouragingly, this marked hinderance of an aggressive rat glioma model was achieved by a single dose of doxorubicin [33] (Table 2).

Cisplatin is a common anticancer drug used to treat extracranial cancers. Its high plasma protein binding affinity leads to a low bioavailability of 5–10%, which combined with its high toxicity and poor BBB penetration, makes it an ineffective brain cancer treatment without reaching systemically toxic levels in patients [45,46]. A gold nanoparticle-bound Cisplatin complex conjugated to an uptake peptide (AuNP-UP-Cis) displayed enhanced DNA damage, cell apoptosis and tumour growth inhibition relative to free cisplatin in brain tumours in vivo. Despite these promising results, the BBB permeability of the nanoconjugate was still considerably low. Promisingly, combining AuNP-UP-Cis with MRgFUS increased BBB permeability by two-fold in mice (Table 2). Additionally, the therapeutic dosage was lowered from 5 mg/kg to 0.5 mg/kg [47], shifting the dosage to the lower region of Cisplatin’s therapeutic index [48].

Another Cisplatin-ferrying nanoparticle conjugate is a PEGylated polyaspartic acid organic nanoparticle that displayed enhanced BBB and Blood–Tumour Barrier (BTB) permeability when used in conjunction with MRgFUS. After three MRgFUS sessions, a significant 6-fold increase in nanoparticle BBB/BTB traversal was measured in one in vivo glioma model (9 L), and a staggering 26-fold in another model (F98; Table 2). In addition to this, tumour growth in the aggressive F98 cancer model was inhibited by 68% and animal survival was extended by 15%. Although not a curative treatment, Timbie et al. propose this approach as a viable treatment option following surgical resection of brain tumors [49].

MRgFUS gene delivery has also produced noteworthy results when utilising viral vectors as vehicles for transporting genes [50,51,52,53,54]. MRgFUS enables specific BBB disruption at selective target sites, increasing site-specific gene delivery with the minimisation of systemic gene distribution [55]. Upon intravenous injection of an adeno-associated virus (AAV) harbouring a green fluorescent protein (GFP)-encoding gene, microbubble-assisted MRgFUS resulted in GFP transduction in 50–75% of neurons of the right striatum after one administration (Table 2). GFP transduction in the contralateral striatum revealed no expression, demonstrating the selective BBB-altering abilities of MRgFUS. Stable expression of GFP in this region was observed for up to 6 months, with a small population of the study reaching 16 months of ongoing stable GFP expression. Stavarache et al. reported “no evidence of substantial toxicity, tissue injury, or neuronal loss” in the brain. There were signs of early microgliosis and astrocytosis at the target site following MRgFUS that remained for 48 h after administration. Undesirably, GFP was shown to be expressed in the liver. It is noteworthy to consider the weak immunogenic response observed in this study, despite the delivery of a viral vector harbouring an exogenous gene that reached deep brain tissue after sonicating the walls of the brain microvasculature. The high efficacy and low toxicity of this approach to neuronal gene delivery makes it an excellent candidate for gene therapy. Further developments however are required to suppress peripheral gene delivery prior to reaching clinical trials, due to the potentially long-lasting effects of the therapeutic at hand. MRgFUS has aided Trastuzumab BBB/BTB permeability in mice, although improvements were limited [56].

#### 2.2.2. MRgFUS BBB-Modulating Clinical Studies

Recently, a phase I clinical trial was carried out to determine if MRgFUS could induce amyloid-β clearance from the brain parenchyma of five Alzheimer’s patients [57]. No significant clearance of amyloid-β was measured, despite successful opening of the BBB. There was no clinically significant worsening on cognitive scores at three months compared to baseline, suggesting short-term safety within humans; this extends to patients with neurodegenerative diseases that may have an already tumultuous neurological environment.

Another phase I clinical study utilising MRgFUS in the delivery of anticancer drugs across the BBB/BTB was evaluated [58]. Mainprize et al. report no adverse clinical or radiologic events in their study, indicating a safe procedure. In addition to a successful primary outcome, the uptake of doxorubicin and temozolomide in the peritumour tissue was significantly increased in patients from whom tissue extraction was possible. This provides promise for therapeutic applications in a future phase II trial.

MRgFUS BBB modulation was evaluated as a method for enabling treatment of amyotrophic lateral sclerosis (ALS) in a phase I clinical study involving four patients [59]. Gadolinium leakage at the eloquent primary motor cortex target site immediately after sonication was observed in all patients, indicating successful BBB opening. In agreement with previous observations, the procedure was deemed to be well-tolerated, with no serious adverse events reported. The ubiquitous findings reported in these phase I clinical trials agree that MRgFUS-mediated BBB modulation is a reportedly safe therapeutic approach that can be applied to various regions of the brain involved in common CNS diseases.

**Table 2 pharmaceutics-13-01980-t002:** Summary of MRgFUS BBB-modulating methods.

BBB Modulator	BBB Permeability Result	Onset of Action	Time to Recover	Tracer(s) Used	NP Size (nm)	Cell Line/Animal Tested	Administration
MRgFUS [32]	3.8-fold increase of Evans blue dye accumulation in healthy brain tissue 2.1-fold increase of Evans blue dye accumulation in brain tumour tissue 1.7-fold increase in TMZ CSF/plasma ratio	<60 s	N.R	TMZ 194 Da Evans blue dye 961 Da	N/A	7–8 week old Male Fischer 344 rats (180 g)	Intravenous injection
MRgFUS [33]	1.4-fold slowing of 9 L gliosarcoma tumour growth	<120 s	N.R	Liposomal Doxorubicin (LDox) 544 Da	100	Male Sprague Dawley rats (~200 g)	Tail vein injection
MRgFUS [35]	22% increase in striatum permeability 26% increase in hippocampus permeability	<15 min	N.R	GFP-ECNPCs	N/A	Sprague Dawley rats (200–250 g)	Carotid artery injection (stem cells) Tail vein injection (microbubbles)
MRgFUS [47]	1.2-fold increase in free CDDP permeability 2.1-fold increase in AuNP	<24 h	<24 h	Cisplatin (CDDP) 300 Da AuNP-UP-CDDP (9 nm)	9	NSG mice	Tail vein injection (All)
MRgFUS [49]	6-fold increase in Cisplatin across the BTB in 9 L glioma rat model 28-fold increase in Cisplatin across the BTB in F98 glioma rat model	<1 h	<1 h	Cisplatin (CDDP) 300 Da PAA-PEG-CDDP	60	Female Fisher 344 rats (200–220 g)	Tail vein perfusion
MRgFUS [56]	50–75% of neurons transduced with GFP in the right striatum	<2 weeks	<2 weeks	~20 nm Virus GFP plasmid (1–5 MDa)	20	Ten-week-old male Sprague Dawley rats (250–300 g)	Tail vein perfusion
MRgFUS [58]	7.7-fold increase in TMZ in tumour tissue 1.5-fold increase in LDox in tumour tissue	<24 h	<24 h	Liposomal Doxorubicin (LDox) 544 Da TMZ 194 Da	250–2500	Human phase I clinical trial (5 patient population)	Intravenous injection (LDox) Oral administration (TMZ)
MRgFUS [59]	~15% increase in Gadolinium leakage at target site	0 min	<24 h	Gadolinium contrast agent (unspecified) (545–975 Da)	N/A	Human phase I clinical trial (4 patient population)	Intravenous injection

Abbreviation: AuNP (gold nanoparticle), AuNP-UP-CDDP (uptake peptide-coated gold nanoparticles functionalised with Cisplatin), CDDP (cisplatin), CSF (cerebrospinal fluid), GFP-ECNPCs (GFP-expressing embryonic cortical neural progenitor cells), N/A (non-applicable), NP (nanoparticle), N.R (not reported), NSG (NOD scid gamma mouse), PAA-PEG-CDDP (Cisplatin PEGylated polyaspartic acid nanoparticle), TMZ (temozolomide). Note: Every MRgFUS method was microbubble enhanced, unless otherwise stated.

## 3. Small Molecule BBB Modulators

### 3.1. Hyperosmolar Agents

Hyperosmolar agents are well-established TJ modulators that have been utilised as BBB modulators for five decades [25]. Dehydration of cerebrovascular endothelial cells leads to their contraction (Figure 3), and this cellular shrinkage results in the opening of TJ gaps, enabling enhanced and indiscriminate paracellular flow of solutes [1,60]. Intracarotid injection of Methotrexate following mannitol injection has been shown to result in a 100-fold increase in delivery of the chemotherapeutic agent to the brain in a canine population [61]. To disrupt the human BBB, a mannitol dose of typically 200 to 300 mL (20–25% *v*/*v* aq solution), injected within a 30 to 40 s timeframe is required [62]. BBB disruption triggered by mannitol has been reported to last between 10 and 40 min for primates (incl. humans) [63,64,65]. Restoration of the BBB back to basal levels after mannitol-induced disruption has been reported to take up to 2 h [66]. Despite this extensive length of time, the barrier is only diminished enough to facilitate non-selective paracellular diffusion for a fraction of this time. Interestingly, due to the large intercellular gaps produced by mannitol BBB disruption, its implementation in brain-targeted stem cell therapy has been investigated. In fact, the combination of stem cell therapy and mannitol has been purported to “improve therapeutic outcomes in adult stroke and neonatal cerebral palsy” [67].

The toxicity of mannitol has been contested, with some studies reporting seizures among patients in addition to adverse cardiac effects [68]. Amelioration of these effects however were subsided by co-administration with relevant medications [69]. In agreement with this, a phase I clinical trial involving co-administration of mannitol and a carboplatin, etoposide and melphalan (CEM) cocktail revealed significant tumour reduction and growth inhibition, with manageable adverse effects being reported [70]. This controlled study provides a strong foundation for further clinical development of mannitol as a BBB disruptor. Furthermore, mannitol induced BBBD-facilitated passage of all anticancer agents of the CEM cocktail, supporting the approach.

As mannitol is indiscriminate in its BBB disruption, widescale neural distribution of therapeutics can lead to systemic neurotoxicity in medications of considerate toxicity. Chemotherapeutics such as cisplatin, bleomycin, 5-fluorouracil, mitomycin, and vincristine are acutely neurotoxic and are incompatible with mannitol-based BBBD, while less neurotoxic agents such as cyclophosphamide, methotrexate, and carboplatin may be utilised in conjunction [71,72].

Noteworthy is the generally inconsistent duration of barrier disruption reported by researchers. For example, Joshi et al. observed that barrier opening duration was inconsistent even within one subject; rabbit brain vasculature opened up for various periods of time at different sites, as observed by optical measurements [73]. On average, the BBB was disrupted for up to one hour in this study which is a similar length reported by numerous other studies [65,66]. Another study reported that rodents had BBBD lasting 6 h following mannitol injection [69]. This inconsistency may benefit from integration of real-time guided administration of mannitol to curb the broad effects of mannitol BBBD. Such a system could be optical feedback, as supported by Kiviniemi et al. [74]; however, implementation of this system relinquishes the low instrumental dependency that gives small molecule BBBD its advantage.

Although mannitol’s use as a BBB modulator was initially reported five decades ago, its presence in this area remains relevant and continues to be implemented with cutting edge therapies such as gene silencing [75] and nanotechnology [76]. Recently, hydrophobically modified small interfering RNA (hsiRNA) was shown to enhance the cellular uptake of siRNA in neuronal cultures without affecting RNA-induced silencing complex (RISC) loading and mRNA silencing [77]. Furthermore, hsiRNA was shown to possess enhanced blood circulating times and reduced renal clearance when compared to unmodified siRNAs [78]. Conjugation of siRNA with phosphocholine-docosahexanoic acid (PC-DHA) was used to silence mRNA of Huntingtin (htt), the protein whose mutation results in Huntington’s disease [75]. Although the PC-DHA-hsiRNAs had limited BBB permeability [77], co-administration with 25% mannitol by intracarotid injection to the right carotid artery facilitated the passage of the PC-DHA-hsiRNAs to the brain parenchyma [75]. Htt gene expression was silenced by 33–55% across various brain regions 1 week after administration according to the analysis of tissue punch biopsies (Table 3). PC-DHA-hsiRNAs were distributed across the ipsilateral hemisphere, the region of the brain supplied by the right carotid system. Despite the limited gliosis observed by Godinho et al. [75], the study reported no associated neurotoxicity.

To augment the specificity of mannitol BBBD, a method that utilises microcatheters to deliver therapeutics to a specific vascular territory—Superelective Intra-Arterial Infusion (SIACI)—has been developed, enabling the delivery of a high drug concentration at the desired endovascular site [79]. This targeted infusion bypasses the systemic circulation achieved by intravenous or standard intra-arterial administration, reducing toxicity and side effects. SIACI has been used in conjunction with mannitol-induced BBBD to selectively increase vascular permeability in vessels supplying glioblastoma tumours. This approach has resulted in increased Bevacizumab [80,81,82,83,84,85] and Cetuximab levels [86] in glioma tissues, extending progression-free and overall survival time with no significant adverse effects reported. In fact, this approach has revitalised mannitol-BBBD so significantly that 11 clinical trials are currently underway in the area.

Interestingly, arabinose has also been reported to disrupt the BBB [87] in a similar manner to mannitol [88,89,90,91]. In addition to mannitol and arabinose, hyperosmolar solutions such as lactamide, saline, urea and radiographic contrast agents can be used to transiently breach the BBB. Despite these options, mannitol remains the most comprehensively investigated in regard to BBB modulation.

### 3.2. Inflammatory Mediators

Endogenous inflammatory mediators have a marked effect on vascular permeabilisation. Histamine and leukotrienes are particularly well understood in this regard [92,93]. All known histamine receptors are expressed on endothelial cells and play various roles ranging from neurotransmission, inflammation, smooth muscle contraction, dilation of capillaries, chemotaxis, cytokine production and gastric acid secretion, as well as vascular permeability alteration [94,95]. These biological processes are controlled by four histamine receptors; H1, H2, H3 & H4; activation of receptors H1, 2 and 4 has been shown to strengthen the BBB [96,97,98], while H3 activation has been shown to increase permeability by elevating Ca^2+^ levels [99]. Doses of 10 and 100 µM histamine appear to increase BBB permeability through transcellular and paracellular processes, respectively [100,101] (Table 3), whilst a larger 1 mM dose strengthens BBB integrity [102]. Development of highly selective histamine receptor agonists and antagonists in recent years have been reported, and show a more discriminate effect compared with histamine. Compounds such as the H3 receptor inverse agonist BF2649 and partial H4 agonist Clobenpropit have been shown to increase BBB integrity [103] (Table 3). Investigators in this area have focused on developing agents that strengthen BBB integrity, rather than transiently weakening it. Currently, numerous H3 agonists are in development; however, to date, none have been shown to disrupt the BBB [104,105,106,107,108,109,110,111,112,113,114,115]. Although a H3 agonist could potentially modulate the BBB, its involvement in neurotransmission might lead to various neurological side effects.

Leukotrienes are another class of inflammatory mediators that, similar to histamine, have broad biological effects involving immune and inflammatory responses [116,117,118,119]. A subclass within this family of biomolecules known as cysteinyl leukotrienes bind to the G-protein coupled receptors CysLTR_1_ and CysLTR_2_, triggering various effects including pulmonary vasoconstriction and bronchoconstriction [116,120]. Substrates for CysLTR_1_ including LTC_4_, LTD_4_ and LTE_4_ have been shown to disrupt the BBB to varying degrees. LTD_4_ induces brain edema [121] and has been reported to facilitate pentylenetetrazol-induced seizures by inducing BBB dysfunction [122]. Co-administration of a CysLTR_1_ antagonist, Montelukast, results in inhibition of the proinflammatory actions of LTD_4_, strengthening of BBB integrity and a subsequent reduction in seizures [122]. In vitro studies also mentioned that LTC_4_ and LTE_4_ alter the BBB, with selectivity toward increasing ischemic BBB regions [123] and BTB [124] permeability over normal BBB tissue (Table 3).

### 3.3. Alkylglycerols

Although the literature is limited, short chain alkylglycerols have been demonstrated as having a significant effect on BBB permeability, with minimal adverse effects. In 2000, Erdlenbruch et al. reported the effect of alkylglycerols on BBB permeability, whereby intracarotid injection of 1-O-pentylglycerol impressively increased the BBB permeability of Methotrexate (230-fold), Cisplatin (125-fold), Vancomycin (15-fold) and Gentamicin (12-fold) in the ipsilateral hemisphere compared to injection in the absence of alkylglycerols [125]. In addition to this, BBB disruption almost reached basal levels after 3 min, and fully reached basal levels after 15 min, demonstrating transient modulation (Table 3). Further investigations by Erdlenbruch et al. widened the scope of compounds that had increased BBB permeability upon 1-O-pentylglycerol coadministration [126,127]. In these studies, 1-O-pentylglycerol increased the BBB permeability of Erucylphosphocholine, Fluorescein and RB 200-albumin in rats (17, 6.55 and 2.7-fold, respectively; Table 3). The relatively lower permeability of the larger albumin and Vancomycin substrates suggests a certain degree of size selectivity and indicates some retention of barrier integrity rather than a total loss of junctional structure. Accumulation of FITC-40 kDa at the luminal surface of endothelial cells suggests that the delivery occurs paracellularly rather than transcellularly, as no vesicular uptake or significant levels of intracellular fluorescence were observed [127]. Investigations into the biodistribution and pharmacokinetic properties of 1-O-pentylglycerol in vivo revealed no significant bioaccumulation, with efficient renal elimination [128]. Using fluorescence-based optical imaging, the spatial distribution of fluorescently-tagged proteins co-administered with 2-O-hexyldiglycerol was visualised in mice. Optical imaging revealed an approximate 1.9-fold increase in fluorescently-tagged globulin (~150 kDa) in the ipsilateral region of the brain [129] (Table 3).

Investigating alkylglycerol’s mechanism of action, Hülper et al. found cytoplasmic redistribution and internalisation of both claudin-5 and β-catenin upon exposure to both 1-O-pentylglycerol and 2-O-hexyldiglycerol, as revealed by immunohistochemistry [129]. Since increased paracellular transport is the suggested mechanism of alkylglycerol treatment, the effect of alkylglycerols on homophilic claudin-5 interactions was determined. Claudin-5 is a key TJ protein that is reported to be responsible for the BBB’s heightened barrier integrity [130]. Hülper et al. reported no significant change in the trans-homophilic interactions of claudin-5, suggesting that alkylglycerols do not interact directly with TJ proteins [131]. Redistribution of junctional proteins and alterations in cell shape indicate the involvement of the cytoskeleton, whereby alkylglycerols may trigger a rearrangement resulting in TJ protein internalisation and transient opening of the BBB [131] (Figure 4); however, further detailed investigations are needed to ascertain a comprehensive understanding of the mechanisms of action. The low toxicity and high permeabilising effects produced by alkylglycerols have spurred studies investigating their incorporation into nanoparticles for use as a BBB shuttle in recent years [132,133]. Despite these developments, subsequent alkylglycerol BBBD research has been scarce.

### 3.4. Sodium Caprate (C10)

Sodium Caprate (C10), is a medium-chain fatty acid (Table 4) that increases the absorption of orally administered drugs across the epithelial and endothelial cell layers [134]. Sodium caprate is already an approved intestinal absorption enhancer for aiding antibiotic permeabilisation in Swedish [135] and Japanese [136] markets and is also a food additive with Generally Regarded As Safe (GRAS) status [137]. Ohnishi et al. first reported the BBB-modulating abilities of sodium caprate in rats and discovered that intraarterial injection of 10 mM sodium caprate resulted in increased flux of a range of substrates into the CNS. The substrates used ranged from 180 Da to 70 kDa and the permeation rate was proportionate to molecular size (Table 3). Ohnishi et al. have hypothesised that BBBD results in paracellular transport through intercellular gaps, as vesicular transport would not have a linear relationship between molecular mass and permeation flux until much larger substrates were used [138].

The BBBD of sodium caprate was confirmed by Preston et al., whereby 15–25 mM modulated the BBB reversibly for 1 h, increasing mannitol (tracer) flux 10-fold—levels similar to hyperosmolar and alkylglycerols agents (Table 3). However, sodium caprate at this concentration also induced high blood pressure in some subjects, which greatly reduced mannitol (tracer) permeability, creating inconsistencies among the subject population [139].

Coadministration of an anti-hypertensive agent alleviated the high blood pressure, removing mannitol flux inconsistencies among the subject population [139]. Sodium caprate was also shown to increase the arachnoid barrier permeability of local anaesthetic ropivacaine by 1.6-fold upon epidural administration, highlighting sodium caprate as an efficient and robust barrier disruptor [140].

Mechanistic insight was provided by Del Vecchio et al., whereby claudin-5 trans interactions were decreased, coupled with a 61% claudin-5 reduction at cell contacts and F-actin internalisation in MDCK-II cells [134]. Immunostaining in bEnd.5 cells also highlighted the loss of expression of both claudin-5 and F-actin at perijunctional membranes. Sodium caprate-induced reorganisation of TJ components resulted in a 2.6-fold increase in lucifer yellow flux (457 Da) [134] (Table 3). Zonula occludens-1 (ZO-1) localisation remained unaffected following exposure to sodium caprate. Considering the preceding mechanistic insights provided by Ohnishi et al. [138], larger molecules should also penetrate the disrupted barrier, but this has yet to be investigated. One would expect based on recent work that claudin-5 down-regulation would lead to the passage of molecules <800 Da [141]; however, since sodium caprate facilitates significant flux for molecules >800 Da, it is likely that more TJ proteins, including JAMs, occludin and/or other claudins, are involved.

### 3.5. Regadenoson

Regadenoson, an A_2A_ adenosine receptor agonist used as a vasodilator during pharmacologic stress tests, has been reported to increase BBB permeability by inhibiting the predominantly expressed [142] efflux transporters P-gp and BCRP in mice [143,144], as well as inducing changes in cytoskeletal organisation and cell shape [145] (Figure 4). Both studies found it to be a fast-acting, quickly reversible and a potent functional inhibitor in mice, increasing the CNS permeability of a diverse range of differently sized molecules [143,144] (Table 3). Despite the promising data reported, two pre-clinical human studies carried out by Jackson et al. revealed no significant change in temozolomide permeability across the BBB upon co-administration with regadenoson (approx. 5 μg/kg) [146,147]. Regadenoson animal studies have reported significant increases in both small and large molecules across the BBB at doses reaching 10-fold less compared to Jackson et al.’s human studies, indicating a possible lack of dosing translatability between animals and humans—possibly due to differences in expression and/or function of BMEC adenosine A_2A_ receptors [143,146,148]. Although the dosing regimen used in animal and human studies may not be comparable, the therapeutic index of regadenoson is wide and can therefore facilitate administration of higher doses. Some animal studies have reported the use of doses reaching 50 μg/kg, with excellent BBB modulation and passage of small and large molecules alike, and have reported no toxic effects [143,144]. Importantly, a clinical trial began in late 2019 that will probe the possibility of administering higher dosages than those already approved by the FDA in indications for vasodilation, to see if BBB permeability can be effectively altered while remaining safe in humans [149] (Table 5).

### 3.6. Fingolimod

A recent study identified the sphingosine 1–phosphate receptor 1 (S1P1) in endothelial cells as a target for BBB modulation. S1P receptors are G protein-coupled receptors (GPCRs), that regulate cell migration, adhesion, survival and proliferation [150]. Endothelial-specific knockout of the S1P1 receptor in mice resulted in BBB leakiness that facilitated a 5-fold increase in the passage of a 3 kDa tracer molecule, whilse no significant alteration in flux was observed for a 10 kDa tracer; i.e., size-selective barrier modulation via S1P1 receptor inhibition [151]. Following this observation, administration of the broad-spectrum S1P receptor inhibitor fingolimod by Yanagida et al. [151] in WT mice resembled the KO study, whereby a 1 kDa tracer had an approximately 3-fold increase in BBB permeability, while the 10 kDa tracer showed no significant enhancement (Table 3). Gilenya^®^, the brand name for fingolimod developed by Novartis, is indicated for the treatment of MS by attenuating peripheral cell trafficking of circulating autoreactive cells into the brain parenchyma [152]. Interestingly an alternative S1P1-selective and potent drug candidate, NIBR-0213, was comparable in terms of BBB modulation to fingolimod, with permeation effects reversing within 48 h (Table 3) [151].

The size-selective BBB opening observed by Yanagida et al. is similar to that previously reported by Keaney et al., where both claudin-5 and occludin were knocked out [153]; however, no significant change in mRNA expression of claudin-5, occludin, ZO-1 or V-cadherin was demonstrated. Interestingly, there was no observable change in TJ structures either [151]. The only detectable alteration was a shift in claudin-5 and occludin from the cytoskeleton to the membrane (Figure 4). Relative to other TJ modulators, S1P1 antagonism appears to subtly alter TJ protein localisation—the exact mechanism of which is currently unknown. Despite this mechanistic uncertainty, Yanagida et al. propose a few possibilities, all relating to actin maintenance and formation [151], as supported by the preceding literature [154]. Two other receptors: lipolysis-stimulated lipoprotein receptor [155] and GPR116 [156], were also recently shown to regulate BBB permeability in a size-selective manner. Similarly to S1P1, no major alterations in TJ morphology were measured in these studies; Yanagida et al. suggest that it is possible that S1P1 may regulate the BBB by co-operating with these receptors. In particular, GPR116 is enriched in endothelial cells and could act alongside S1P1 through common mechanisms [151].

### 3.7. NS1619

NS1619, a BK channel activator, has been shown to induce a 4-fold increase in paracellular flux of a 44 kDa tracer across the blood–tumour barrier (BTB) by downregulating expression of both claudin-5 and occludin [157] (Table 3). Biochemical investigations into the mechanism of action reveal transient activation of the PI3 kinase and Akt pathways via ROS/RhoA [157]. As this agent facilitates the delivery of macromolecules to cancer cells across BMECs in vitro, it holds promise for further investigation in animal studies.

### 3.8. NEO100

Wang et al. have recently described the intra-arterial injection of NEO100—a highly purified monoterpene (S)-perillyl alcohol, which has been shown to reversibly and dose-dependently open the BBB without toxicity in vivo [158]. Full recovery of the barrier was observed four hours following the administration of NEO100, which induced translocation of tight junction proteins to the cytoplasm in brain endothelial cells. Interestingly, Neonc Technologies are currently investigating the safety, pharmacokinetics, and efficacy of a repeated dose regimen of NEO100 delivered intranasally for the treatment of patients with recurrent glioblastoma in a US-based phase 1/2a clinical trial [159]. This trial was preceeded by a phase 1/2 clinical trial using a lower grade perillyl alcohol which indicated improved survival for a patient group in Brazil [160].

### 3.9. M01

In a recent publication, Breitkreuz-Korff et al. reported M01; a derivative of Nalidixic acid, having a potent effect on BBB modulation [161]. Intravenous tail injection of M01 formulated within a cyclodextrin resulted in a 3.9-fold increase in BBB permeation of a fluorescein tracer (Table 3). Administration of the chemotherapeutic paclitaxel in conjunction with this modulator over a 4-week period resulted in a 75% reduction in tumor area within an in vivo mouse model. As M01 targets the extracellular domain of claudin-5, a tight junction protein found primarily within the paracellular gaps of endothelial cells [162], it provides a degree of selectivity as indicated by the lack of increased fluorescein levels within proximal organ tissues. The extracellular domain of claudin-5 has been targeted by peptides in recent years, and while these agents show promising efficacy as BBB modulators, they inherently possess a greater tendency for degradation and rapid elimination relative to small molecules such as M01, leading to a shorter bioavailability unless formulated or modified correctly [163].

## 4. Peptide and Peptidomimetic BBB Modulators

### 4.1. RMP-7

A common effect observed by inflammatory mediators is their ability to increase BBB permeability temporarily (histamine [164], bradykinin [165] and leukotriene [166]). Considering this, it is clear why the development of synthetic derivatives of these biomolecules has been an area of interest in CNS drug delivery. Bradykinin is an endogenous inflammatory mediator that targets bradykinin receptors B_1_ and B_2_. B_1_ expression is activated in response to chronic pain and inflammation, whilst B_2_ is constitutively expressed and mediates vasodilation and tissue permeability [167]. Bradykinin receptor B_2_ is a GPCR that, upon activation, results in a reduction in cAMP levels, upregulation of myosin light-chain kinase phosphorylation with subsequent opening of TJs by actin–myosin contraction and the formation of stress fibers [168,169] (Figure 5). Alkermes developed RMP-7 (Cereport)—a peptide derivative of bradykinin that targets bradykinin receptor B_2_, maintains an enhanced plasma half-life relative to bradykinin and exhibits efficacy as a BBB modulator [170].

Numerous publications have revealed that RMP-7 increases carboplatin BBB permeability [171] with additional affinity for brain tumour tissues [172]. The permeability achieved by RMP-7 was not size-specific however, with permeation of 70 kDa dextran reaching a 4-fold permeation rate relative to the control [173] (Table 3). Despite side effects being observed, numerous phase I clinical trials combining RMP-7 and Carboplatin have shown good tolerability among patients [174,175,176]. Unfortunately, subsequent phase II studies have failed to show statistically significant effects on high-grade gliomas [177,178,179,180]. Despite these failures, there was criticism over certain aspects involving the dosing regimen, which had they been modified, may have resulted in improved outcomes [167].

### 4.2. Zonula Toxin and Analogues

Zonula toxin (Zot) is a 45 kDa protein produced by the pathogenic bacterium *Vibrio cholerae* [181]. Zot increases TJ permeability in both epithelial and endothelial cells and has been implicated in the pathogenesis of coeliac disease and type 1 diabetes [182]. Zot is an integral membrane protein with an extracellular region, that upon cleavage, is released as a 12 kDa carboxyl-terminal peptide [183], which is proposed to interact with the proteinase-activated receptor 2 (Par-2), leading to protein kinase c-dependent actin reorganisation and TJ modulation of barrier permeability [184]. The 12 kDa fragment of Zot in both epithelial [185] and endothelial cells increased the permeability of sucrose, doxorubicin, and paclitaxel by two-fold (P_app_) [186] (Table 3).

Developing Zot further, the 12 kDa biologically active fragment, DeltaG (∆G), was isolated [187], and was shown to lead to a 2.5 and 7-fold in vivo BBB permeability increase in paclitaxel and methotrexate brain distribution, respectively [188] (Table 3). Further simplification of Zot’s structure in an attempt to identify the key motif responsible for its TJ-modulating activation was investigated by Song et al. [189]. Site-directed mutagenesis, in combination with sequence homology analysis between ∆G and Zot, revealed an octapeptide region responsible for Zot’s activity. This region was once again simplified to generate a hexapeptide that is responsible for Zot’s TJ modulation, known as AT-1002 [189].

The refinement of Zot to a hexapeptide simplifies ease of synthesis; unfortunately however, potency was reduced by five orders of magnitude [190] (Table 3). Interestingly, AT-1002 was shown to undergo dimerisation in relevant biological systems via P2 cysteine bridge formation, dramatically hindering the efficacy of the hexapeptide [191]. In Li et al.’s structure–activity relationship (SAR) investigation, substitution of the cysteine side chain residue to an allyl group removed the peptides dimerisability and restored its effects of altering intestinal permeability [191] (Table 3).

Investigations into the mechanism of action of AT-1002 have revealed that the hexapeptide bears homology resembling a PAR_2_-activating motif. Subsequent PAR-2 activation would increase PKCα-dependent ZO-1 and myosin 1C serine/threonine phosphorylation, inducing selective disassociation of ZO-1 from its TJ protein-binding partners (occludin, claudins, JAM’s) and myosin 1C [190] (Figure 5). Interestingly, a structurally similar octapeptide known as larazotide, being developed by 9 Meters Biopharma as an adjunctive therapy for the treatment of coeliac disease, has been shown to attenuate intestinal barrier disruption—producing the opposite effects of AT-1002. Its promising bioactivity has resulted in it reaching phase IIb clinical trials as a potential treatment for coeliac disease [192].

### 4.3. PN-159

PN-159 is an 18-mer, polycationic, amphipathic peptide discovered by screening of a custom peptide library, with members possessing varying degrees of helicity and hydropathy [193] that has exhibited promise as a TJ modulator of epithelial cells, reaching phase II clinical trials for the treatment of weight loss and nasal administration of insulin [167]. Its efficacy as a BBB modulator has remained untested until recently. Bocsik et al. compared the efficacy of a set of TJ-modulating peptides for modulating intestinal and endothelial barriers in vitro [194]. While all peptides altered the transendothelial electrical resistance (TEER) of both barriers significantly (TEER reduction approximately 30–60% for the intestinal and 35–60% for the BBB model), PN-159 displayed best-in-class results with 96% and 70% TEER reduction for intestinal and BBB models, respectively. In addition to TEER alterations, PN-159 enabled an 11 and 9.5-fold increase in fluorescein and albumin flux in the BBB model, respectively. The second largest BBB flux level measured was from AT-1002 administration, followed by 7-mer and then ADT-6, which had marginal changes in flux (Table 3). All cells returned to basal TEER levels within 24 h, demonstrating reversibility as BBB modulating agents. Immunostaining and affinity measurements revealed some insight into PN-159’s mechanism of action. PN-159 caused widespread internalisation of claudin-5 with decreased ZO-1 expression and β-catenin at the cell borders of primary brain microvascular endothelial cells (Figure 5). Microscale thermophoresis measurements determined that PN-159 had a binding affinity to claudin-1 and 5 of 820 nM and 327 nM, respectively [194].

### 4.4. HAV-6, C-CPE and Their Derivatives

Despite both HAV-6 and C-CPE having altered endothelial TEER levels, Bocsik et al. [194] observed no significant alterations in the BBB permeability towards small or large molecular tracers. ADT-6 had a slight effect on tracer flux; however, it was minimal (Table 3). Contrastingly, Ulapane et al. carried out an in vivo study with HAV-6, cHAVc3 and ADTC5—the latter two being cyclic derivatives of HAV-6 and ADT-6, respectively. Administration of these peptides resulted in a fast (10–40 min), reversible opening of the BBB, which facilitated a significant (2 to 4.8-fold) increase in peptide and protein tracer passage (775 Da–65 kDa, Table 3) [195]. NMR binding and molecular docking studies suggested that HAV-6 and cHAVc3 interrupt cadherin cis-interactions, whilst ADTC5 interrupted cadherin trans-interactions—a conclusion supported by previous studies [196].

C-CPE’s weak BBB modulation, reported by Bocsik et al., is explained by its affinity for claudin-3 and 4—two claudins that are expressed on epithelial cells; however, not particularly in the context of BMECs [194,197]. C-CPE mutants designed to bind to claudin-5 were generated by Protze et al. using structure-based mutagenesis [198]. Of the various mutants, Neuhaus et al. found C-CPE-Y306W/S313H exhibited a 68% drop in TEER and a 1.9-fold increase in tracer flux (376 Da) [199] (Table 3). Liao et al. further investigated this claudin-5 specific C-CPE ligand in an in vitro and in vivo study. Greater flux of a larger tracer (~4.7 fold, 10 kDa) was measured by Liao et al. in zebrafish larvae, with a shorter BBB opening time reported (3–4 h) [200] (Table 3). While both studies indicate that this C-CPE mutant opens the BBB, the extent to which it does so remains disputed.

### 4.5. Claudin Extracellular Loop Mimics

Claudin-1 extracellular loops (ECLs’) mimicking the second half of the first ECL have shown barrier opening effects in epithelial and endothelial cells [201,202]. Peptide C1C2 caused a 50% TEER reduction and an 8.25-fold increase in the flux of both Luciferase yellow (444 Da) and AlexaFluor 680-dextran (3 kDa) in primary mouse BMECs 24 h after C1C2 addition [202] (Table 3). Investigations into the mechanism of action of C1C2 have revealed that it triggers the cytosolic internalisation of occludin and claudins 1 to 5 [202]. Of these claudins, only claudin 1 and 5 have shown direct interactions with one another [203]. Staat et al. speculate that C1C2 disrupts claudin-1 and 5 directly and that their resultant removal from cell borders diminishes other TJ proteins that are involved in heterophilic interactions with them—initiating a domino effect of mislocalisations at TJs. C1C2 drastically altered claudin-5 morphology from discontinuous E-face to continuous P-face-associated strands along the cell border, as shown by freeze fracture experiments [202]. This was observed previously in single point mutagenesis experiments of claudin-5 [204]. The residues that caused this structural transformation also hindered claudin-5 trans interactions. Based on this data, it is suspected that C1C2 interacts with all key ECL amino acids that cause drastic morphological alterations at the cell border upon their mutation. Internalisation of C1C2 in recyclosomes via a clathrin-mediated pathway was observed in vitro and is speculated to explain the delayed onset of action of its TJ-altering effects [205] (Figure 5). C1C2-mediated TJ modification was applied to increase the permeability of hydrophilic antinociceptive agents upon perineurial injection in rats [206]. This study investigated these effects in epithelial tissue, although comparable results might be expected in endothelial tissue based on C1C2’s established effect on the endothelium. In addition to having BBB-permeabilising properties, C1C2 also shows promising barrier repairing properties in stroke patients [207]. Post-stroke patients have shown a subsequent upregulation of claudin-1 and a downregulation of claudin-5 at the BBB. This alteration in claudin expression results in a leaky BBB, leading to a vulnerable brain parenchymal microenvironment. Sequestration of junctional claudin-1 by C1C2 facilitates upregulation of claudin-5 and its return to basal levels—restoring the BBB of post-ischemic stroke patients. C1C2 thus demonstrates significant potential as a therapeutic for a range of CNS diseases [207].

Dithmer et al. developed a claudin-5-targeting peptide mimicking the first ECL loop of claudin-5 which displayed BBB modulating activity [208]. C5C2 decreased TEER by 40% and increased the flux of a wide range of tracers significantly (457 Da–67 kDa). Inversion of all of C5C2’s *α*-carbon stereochemical centers from L to D-configurations (D-aa-C5C2) created a more efficacious and pharmacokinetically stable peptide [208]. An in vivo study in rats revealed that D-aa-C5C2 had reduced peptidase activity, favouring a longer half-life and clearance time. Exposure of D-aa-C5C2 to bEND.3 cells caused a 30% decrease in membranous claudin-5 and a 34% increase in cytosolic claudin-5. Additionally, ZO-1 and occludin were downregulated, based on total cell mRNA levels. Accompanied by this reorganisation of TJ and scaffolding proteins, the cell morphology shifted to a more rounded and shortened shape [208]. Due to the similar peptide design and effects observed with C1C2 by Staat et al. [202], a similar mechanism for this peptide is suggested; however, further investigations are required to confirm this claim.

### 4.6. Occludin ECLs—Epithelial Disruptors

A synthetic peptide based on the second extracellular loop of chicken occludin (OCC2) has been shown to reversibly disrupt the transepithelial permeability of the barrier in Xenopus kidney epithelial cells. A TEER reduction of ≈6000 ΩVcm^2^ to ≈900 Ωcm^2^ was observed following OCC2 administration, while no significant TEER change was observed for OCC1—a synthetic peptide based on the first extracellular domain of occludin [209]. OCC2 resulted in a ubiquitous 10-fold increase in the paracellular flux of mannitol (184 Da), inulin (5.2 kDa), dextran 3K (3 kDa), and dextran 40 K (40 kDa; Table 3). Furthermore, no changes in TJ morphology or localisation of other TJ proteins were observed [209]. Conversely, OCC1 has also been shown to exhibit barrier-modulating effects in the Caco-2 cell line, while OCC2 provided little to none in a subsequent study [210]. Interestingly, simplification of the peptide sequence of OCC1 (OP90–135) resulted in a jump in mannitol flux from 3-fold to 45-fold (OP90–113). The variance between the two studies can be explained by their differing sequences, with Wong et Gumbiner having synthesised occludin peptides based on the chicken occludin sequence, while Tavelin et al.’s proceeding study was based on the human occludin sequence [210,211]. Despite these promising results, no significant data investigating the effect of either occludin peptide on BMEC permeability has been published. As occludin is highly expressed at the BBB [212], the selective OCC1 might be an ideal candidate for human BBB modulation.

## 5. Protein BBB Modulators

### 5.1. Angubindin-1

Deviating from targeting proteins found at bicellular TJs (bTJs), Angubindin-1, a section from the Ib domain of the *Clostridium perfringens* iota-toxin (amino acids 421–664) [213] was reported by Zeniya et al. to target tricellular TJs (tTJs) [214]. Angubindin-1 binds to angulin-1 and 3;—two components that each, in association with tricellulin, form tTJs [214]. Since tTJs are not distributed widely nor frequently enough to facilitate efficient ion permeability, combined with their larger junctional area relative to bTJs, Zeniya et al. proposed that disruption of these tricellular contact points may result in efficient macromolecular paracellular flux. Angubindin-1 disrupted tTJs in mice, resulting in a 20-fold increase in antisense oligonucleotide (ASO) levels (5.3 kDa), with no overt adverse effects observed (Table 3). Of note, total barrier breakdown was not induced, as albumin (67 kDa) failed to enter the CNS, indicating a size-selective effect—although the approximate cut-off is not known due to the lack of inclusion of a range of size-selective tracers in the study. One particularly attractive feature of this approach was the limited increase in the epithelial permeability of peripheral organs. The preliminary results, in terms of both efficacy and toxicity, reported by Zeniya et al. highlight targeting of tricellular contact points as a viable alternative approach for modulating the BBB, and should stimulate additional research in the area.

### 5.2. Gintonin

Gintonin, an extract from ginseng, was reported to rapidly modulate the BBB and grant high BBB traversal for macromolecules between 10–70 kDa [215]. An in vitro BBB assay by Kim et al. showed a rapid and large (<1 min, ~110-fold) increase in Texas red Dextran passage (70 kDa). Follow-up in vivo assays showed a reduction (<5 min, 4-fold) in tracer (10 kDa) crossing to the brain (Table 3). The rather large dose (10 mg/kg) of gintonin required to bring about these desirable modulative properties is a drawback; however, identifying and isolating the active component(s) might facilitate a reduction in dose whilst maintaining efficacy.

Mechanistic investigations into Gintonin’s effect at the BBB have revealed that the active component(s) bind to lysophosphatidic acid receptors 1 and 3 (LPA-1/3). Their activation, along with downstream ROCK pathway activity, induces morphological changes and subsequent redistribution of VE-cadherin, claudin-5, occludin and ZO-1 away from TJs; creating temporarily vulnerable paracellular gaps [215]. Despite the ambiguity as to the exact component(s) responsible for Gintonin’s BBB modulative effects, Ginseng major latex-like protein 151 (GLP151) is a major constituent of Gintonin that binds to LPA [216] and might serve as the likely effector; however, further evidence is required to confirm this.

### 5.3. Antibodies

Similar to the approach of targeting claudin-5 via synthetic ECL peptides taken by Staat et al. [202], Hashimoto et al. developed a range of anti-claudin-5 monoclonal antibodies (mAbs) with affinities for various regions of claudin-5’s extracellular loops [217]. Treatment with two mAbs in particular (M48 and R9), caused the internalisation of claudin-5 from TJs to the cytosol, resulting in a 3 to 3.75-fold increase in fluorescein (376 Da) and a 3.5 to 5.75-fold increase in fluorescein-dextran (4 kDa) in vitro, respectively (Table 3) [217]. Epitope mapping by Hashimoto et al. revealed that both M48 and R9 bind to ECL2 of claudin-5 [217]—the loop involved in claudin-5 trans interactions [204]. The exceptionally high expression of claudin-5 at the BBB is potentially very beneficial, as a claudin-5-specific modulator will have a selective effect on the BBB endothelium, hindering its side effects in peripheral tissues [217,218,219]. Importantly, Hashimoto et al. also demonstrated no significant drop in TEER upon exposure to either mAb in epithelial cell permeability assays using Caco-2 and T84 cell lines [217].

The high specificity of antibodies has also been applied to transcellular shuttling of therapeutics across the BBB. Iduronate 2-sulfatase (IDS), a therapeutic enzyme used to treat lysosomal storage diseases, was incorporated by Ullman et al. into the FC domain of human Immunoglobulin G1 (IgG1) [220]. Conjugation of IDS with the IgG1 FC domain provided affinity for the transferrin receptor (TfR) [221], enabling the efficient transport of this therapeutic across the BBB, resulting in the supplementation of IDS-deficient neurons, astrocytes and microglia. This particular approach has gained significant recognition [222] and holds promise for an efficient and selective method of traversing the BBB.

## 6. Oligonucleotide-Based Gene Silencing

### 6.1. RNAi

Claudin-5 silencing by siRNA has been shown to enhance BBB permeability towards biotin (600Da) [223] in addition to the bioimaging agent Gd-DTPA (742 Da) [224], with effects lasting up to 48 h. Additionally, FITC dextran (4400 Da) failed to permeate the BBB upon claudin-5 knockdown, demonstrating the size-selective properties of claudin-5 [224] (Table 3). In a similar study, Van Itallie et al. demonstrated the key role the scaffolding protein ZO-1 plays in the barrier formation of epithelial cells by depleting its expression through siRNA [225]. ZO-1 silencing in MDCK cells caused a 2-fold permeability increase in molecules ranging from 180 Da to 3 kDa (Table 3). As ZO-1 facilitates the correct localisation of almost all TJ proteins at the tight junction, silencing of this key scaffolding protein may weaken the TJs beyond repair. Interestingly, dual siRNA knockdown of claudin-5 and occludin resulted in a paracellular diffusion of amyloid-β peptides, enabling the clearance of amyloid-β aggregates within the brains of mice, and subsequent improvement of cognitive function [153] (Table 3). The similarity in BBB alteration upon either ZO-1 or claudin-5 and occludin silencing suggests these TJ proteins play primary roles in regulating small molecule BBB permeability.

More recently, siRNA conjugates were shown to penetrate brain microvascular endothelial cells in vitro. A block copolymeric nanoparticle coated with a transferrin receptor peptide substrate to facilitate receptor-mediated transcytosis encapsulated P-glycoprotein (P-gp) siRNA and reduced P-gp expression by up to 52%. The resultant curbing of P-gp efflux enhanced the BBB permeability of rhodamine 123 by up to 27% relative to standard P-gp activity, 92 h after cell treatment [226] (Table 3). Verapamil, a small molecule inhibitor of P-gp, was used as a positive control, enhancing rhodamine 123 permeability by 49% relative to standard P-gp activity. This nanostructure shows great promise for siRNA treatment; however, further developments will need to be made.

### 6.2. Antisense Therapy

Antisense oligonucleotides represent a novel class of synthetic nucleic acid molecules that are capable of bringing about changes in gene expression, as detailed by Rinaldi et Wood [227]. In this regard, Kuwahara et al. developed a 13-mer heteroduplex oligonucleotide that serves as a gene silencer for the transporter OAT3 [228]. Heteroduplex oligonucleotides are comprised of a DNA strand that has locked nucleotides capping either end of the DNA, forming a gapmer. This gapmer is complexed with an siRNA active complementary RNA strand (cRNA), creating the heteroduplex oligonucleotide (HDO). The backbone of either or both strands are usually modified by a 2’-O-methyl modification, and linked by a phosphorothioate substitution. The backbone modification promotes greater stability by reducing susceptibility to nucleases and increasing binding affinity to serum proteins—hindering kidney excretion and clearance [229]. Within the target cell, the DNA strand is susceptible to RNAase H cleavage, causing release of the active RNA strand. The RNA strand targets the complementary mRNA within the nucleus encoding a target protein (Figure 6). Heteroduplex oligonucleotides provide the gene silencing abilities of RNAi with the stability of DNA constructs. The HDO complex was conjugated to the delivery molecule α-tocopherol (Toc-HDO) and was intravenously injected without any additional delivery vectors. The addition of the α-tocopherol increased the hydrophobicity of the construct, enabling greater cell permeability, which is speculated to be mediated by the low-density lipoprotein receptor (LDLR)10, as α-tocopherol is also a substrate of this RMT transporter. Seventy-two hours after administering Toc-HDO, the OAT3-mediated transport rate of radioactive brain imaging agent 99mTc-ECD was lowered by 55% [228] (Table 3).

**Table 3 pharmaceutics-13-01980-t003:** Summary of BBB modulators.

BBB Modulator	BBB Permeability Result	Onset of Action	Time to Recover	Tracer(s) Used	Cell Line/Animal Tested	Administration
Small Molecules						
Hyperosmolar Agents						
Mannitol ^a^ (45 mL, 25% *w*/*v*) [61]	100-fold increase of MTX in Brain tissue	<30 min	N.R	Methotrexate (MTX) 454 Da	Adult mongrel dogs (20–25 kg)	Internal carotid artery injection
Mannitol ^a^ (45 mL, 25% *w*/*v*) [61]	10-fold increase of MTX in Brain tissue	<30 min	N.R	Methotrexate (MTX) 454 Da	Adult mongrel dogs (20–25 kg)	Femoral vein injection
Mannitol (180–360 mL, 25% *w*/*v*) [63]	1000% increase in BBB permeability 60% increase in BTB permeability	4 min (brain) 4 min (tumour)	43 min (brain) 35 min (tumour)	Methotrexate (MTX) 454 Da	Thirteen glioblastoma multiforme patients	Intracarotid injection (All)
Mannitol ^a^ (30 mL, 25% *w*/*v*) [65]	10-fold increase in [^68^Ga]EDTA	30 s	10 min	[^68^Ga]EDTA 356 Da	Five adult rhesus monkeys (5–10 kg)	Intracarotid injection (mannitol) Intravenous (^68^Ga EDTA)
Mannitol ^a^ (30 mL, 25% *w*/*v*) [66]	2.5 increase in influx constant	30 s	2 h (extrapolated)	Rubidium 82 82 Da	Six adult male baboons (25–30 kg)	Intracarotid injection (mannitol) Peripheral intravenous injection (Rubidium 82)
Mannitol ^a^ (8 mL/40 s, 25% *w*/*v*) [73]	5-fold increase in EBD brain accumulation	<40 s	>1 h	Evans Blue Dye (EBD) 961 Da	New Zealand white rabbits	Intracarotid injection (mannitol) Intravenous injection (EBD)
Mannitol ^a,b^ (2.25 mL/25 s, 25% *w*/*v*) [75]	~4 to 55-fold increase in siRNA in brain tissue relative to saline control	<48 h	<48 h	Cy3-PD-hsiRNA ~13–15 kDa	8–12 week male Sprague-Dawley rats (~325 g)	Intracarotid injection (All)
Arabinose ^a,b^ (2 g/Rat) [88]	19-fold increase in brain permeability	<15 min	2 h	[^14^C]Sucrose 342 Da	Male adult Osborn-Mendel strain rats (250–350 g)	Right carotid artery perfusion
Inflammatory Mediators						
Histamine (100 μM) [100]	20% drop in TEER	5 min	>30 min	NTU	Co-culture model: HUVEC-304 C6 glioma cells (12-well)	In vitro
Histamine ^c^ (10 μM) [101]	4-fold increase in Evans blue albumin (EBA) flux	<15 min	>2 h	EBA 67 kDa	Co-culture model: Bovine BCECs Primary rat astrocytes (6 well)	In vitro
Leukotriene D4 ^b^ (6 pmol/Mouse) [122]	1.3-fold increase in brain:serum % of fluorescence marker	<35 min	>35 min	Sodium Fluorescein 355 Da	Adult male Swiss mice (25 ± 3.5 g)	ICV injection
Alkylglycerols						
1-O-pentylglycerol ^a^ (39 mg/Rat) [125]	Increase in tracer permeabilities: Methotrexate (230-fold) Cisplatin (125-fold) Vancomycin (15-fold) Gentamicin (12-fold)	<3 min	15 min	Methotrexate (MTX) 454 Da Cisplatin (CDDP) 300 Da Vancomycin (VCM) 1449 Da Gentamicin (GTM) 478 Da	Male Wistar rats (250–320 g)	Right internal carotid artery injection
1-O-pentylglycerol ^d^ (39–57 mg/Rat) [126]	17-fold increase in Erucylphosphocholine (EPC)	<5 min	N.R	EPC 490 Da	Male Wistar rats (230–305 g)	Intracarotid bolus injection
1-O-pentylglycerol ^a,b^ (90 ± 10 mg/kg) [127]	6.5-fold increase in sodium fluorescein 2.7-fold increase in lissamine-rhodamine B200 (RB 200) albumin	<8 min	N.R	Sodium Fluorescein 367 Da RB 200 albumin 70 kDa	Wistar rats (180–220 g)	Intracarotid injection
2-O-hexyldiglycerol ^a,b^ (1.2 mL/18 s, 100 mM) [129]	~1.9-fold increase in RB 200 γ-globulin brain permeability	≤10 min	~24 h	RB 200 γ-globulin ~150 kDa	Wistar rats (180–220 g)	Intracarotid injection
Other						
Sodium Caprate ^b^ (7.5 mM) [134]	~2.6-fold increase in Lucifer Yellow permeability	<10 min	>40 min	Lucifer Yellow 457 Da	Monoculture model: MDCK-II cells (24 well)	In vitro
Sodium Caprate ^a,b^ (20 mM) [138]	Increase in tracer BBB permeabilities: Mannitol 7-fold Sucrose 6.4-fold PEG 900 5.6-fold PEG 4000 3.6-fold FITC dextran 4000 3.3-fold FITC dextran 20,000 3.2-fold FITC dextran 70,000 2.2-fold	2–5 mins	>15 min	Mannitol 182 Da Sucrose 342 Da PEG 900 900 Da PEG 4000 4000 Da FITC dextran 4K 4400 Da FITC dextran 20K 19,600 Da FITC dextran 70K 71,200 Da	Male Wistar rats (200–250 g)	Internal carotid artery perfusion
Sodium Caprate ^a^ (8.7 mg/Rat) [139]	10-fold increase in Mannitol brain permeability	30–90 s	1 h	Mannitol 182 Da	Adult sprague dawley rats (360–380 g)	Left internal carotid artery infusion
Regadenoson ^b^ (0.5 μg/kg) (3 doses, 5 min apart) [143]	Approx. 3-fold increase in dextran BBB permeability	<35 min	35 min	Dextran 10 kDa	Sprague Dawley rats female, 8 weeks (200–220 g)	Retro-orbital intravenous injection
Regadenoson ^b^ (50 μg/kg) [143]	Approx. 4-fold increase in dextran BBB permeability	<35 min	35 min	Dextran 10 kDa	Sprague Dawley rats female, 8 weeks (200–220 g)	Retro-orbital intravenous injection
Regadenoson ^c^ (50 μg/kg) [144]	Approx. 5, 10 and 11-fold increase in epirubicin within the cerebellum, hippocampus and cortex respectively	5–15 min	30 min	Epirubicin 544 Da	Wild type mice (unspecified)	Intravenous injection
Regadenoson (0.5 μg/kg) [148]	60% increase in temozolomide BBB permeability	<1 h	N.R	Temozolomide 194 Da	Female F344 rats (150–170 g)	Intravenous tail injection
Fingolimod ^b^ (5 mg/kg) [151]	2.7-fold increase in Alexa Fluor 555–cadaverine (AFC) leakage	<6 h	<7 days	AFC 1 kDa	Wild type mice	Oral gavage
NIBR-0213 ^b^ (60 mg/kg) [151]	5-fold increase in Alexa Fluor 555–cadaverine (AFC) leakage	<6 h	<48 h	AFC 1 kDa	Wild type mice	Oral gavage
NS1619 ^b^ (10 μM) [157]	40% drop in TEER 4-fold increase in horseradish peroxidase (HRP) flux	1–2 h	4–6 h	HRP (44 kDa)	Co-culture model: Rat BMECs C6 glioma cells (24 well)	In vitro
M01 ^b^ (2.9 µmol/kg) [161]	3.9-fold increase in Fluorescein levels within cerebrum	<3 h	24–48 h	Sodium Fluorescein 367 Da	Adult C57BL/6N mice	Intravenous tail injection
Peptides, Peptidomimetics & Proteins						
RMP-7 ^d^ (1.5 µg/kg) [171]	2.7-fold increase in tumour permeability of carboplatin	<20 min	35–65 min	Carboplatin 373 Da	Female Wistar rats (180–230 g)	Intracarotid infusion (RMP-7)
RMP-7 ^d^ (1.5 µg/kg) [173]	4-fold increase in 70 kDa dextran	<5 min	25 min	Dextran 70k 70 kDa	Wistar rats RG2 glioma model	Intra-arterial infusion
Zonula occluden toxin ^b^ (2 μg/mL) [186]	2-fold increase in sucrose, doxorubicin and paclitaxel across BBB monolayer 1.3-fold increase in insulin across BBB monolayer 32% drop in TEER	30 min	80 min	Sucrose 342 Da Doxorubicin 544 Da Paclitaxel 854 Da Insulin 5734 Da	Bovine BMEC monolayer	In vitro
∆G ^b^ (600 μg ∆G/kg) (MTX) (800 μg ∆G/kg) (PTX) [188]	7-fold increase in brain:plasma ratio (MTX) 2.5 increase in brain:plasma ratio (PTX)	<5 min	N.R	Sucrose 342 Da Methotrexate (MTX) 454 Da Paclitaxel (PTX) 854 Da	Male Sprague–Dawley rats (225–275 g)	Intracarotid cannula
ADT-6 ^b^ (2 mM) [194]	60% reduction in TEER 1.5-fold increase in fluorescein flux No significant increase in albumin flux	<1 h	1–24 h	Fluorescein 376 Da Albumin 65 kDa	Triple Culture BBB: Primary BMECs Glial cells Pericytes (12 well)	In vitro
HAV-6 ^b^ (2 mM) [194]	60% reduction in TEER No significant increase in fluorescein flux No significant increase in albumin flux	<1 h	1–24 h	Fluorescein 376 Da Albumin 65 kDa	Triple Culture BBB: Primary BMECs Glial cells Pericytes (12 well)	In vitro
C-CPE ^b^ (1 mM) [194]	28% reduction in TEER No significant increase in fluorescein flux No significant increase in albumin flux	<1 h	1–24 h	Fluorescein 376 Da Albumin 65 kDa	Triple Culture BBB: Primary BMECs Glial cells Pericytes (12 well)	In vitro
7-mer ^b,c^ (100 μM) [194]	49% reduction in TEER 5.5-fold increase in fluorescein flux 3.5-fold increase in albumin flux	<1 h	1–24 h	Fluorescein 376 Da Albumin 65 kDa	Triple Culture BBB: Primary BMECs Glial cells Pericytes (12 well)	In vitro
AT-1002 ^b^ (2 mM) [194]	48% reduction in TEER 6.5-fold increase in fluorescein flux 6-fold increase in albumin flux	<1 h	1–24 h	Fluorescein 376 Da Albumin 65 kDa	Triple Culture BBB: Primary BMECs Glial cells Pericytes (12 well)	In vitro
PN-159 ^b^ (10 μM) [194]	68% reduction in TEER 11-fold increase in fluorescein flux 9.5-fold increase in albumin flux	<1 h	1–24 h	Fluorescein 376 Da Albumin 65 kDa	Triple Culture BBB: Primary BMECs Glial cells Pericytes (12 well)	In vitro
HAV-6 ^b,e^ (10 μmol/kg) [195]	2.7-fold increase in Galbumin flux (posterior brain) 3.5-fold increase in Galbumin flux (midbrain) 3.2-fold increase in Galbumin flux (anterior brain)	<3 min	<10 min	Galbumin 65 kDa	Balb/c mice	Tail vein injection
ADTC5 ^b,e^ 7.7 mg/kg (Galbumin and IRdye) 30 mg/kg (cIBR7 assay only) [195]	3-fold increase in Galbumin flux (posterior brain) 4.8-fold increase in Galbumin flux (midbrain) 3.5-fold increase in Galbumin flux (anterior brain) 2.8-fold increase in IRdye800cw-cLABL brain to plasma fluorescence 4-fold increase in cIBR7 brain levels	<3 min	<40 min	Galbumin 65 kDa IRdye800cw-cLABL 2.2 kDa cIBR7 775 Da	Balb/c mice (cIBR7 and Galbumin) male Sprague–Dawley rats (300–400 g) (IRdye assay)	jugular vein cannulation (IRdye assay) Tail vein injection (cIBR7 and Galbumin)
cHAVc3 ^b,e^ (6.6 mg/kg) [195]	2-fold increase in Galbumin flux (posterior brain) 4.2-fold increase in Galbumin flux (midbrain) 3.2-fold increase in Galbumin flux (anterior brain)	<3 min	>51 min	Galbumin 65 kDa	Balb/c mice	Tail vein injection
cCPE-Y306W/S313H ^b^ (10 μg/mL) [199]	1.9-fold increase in Carboxyfluorecein flux 68% drop in TEER	<5 h	35 h	CF 376 Da	Monoculture BBB: pPBMEC (12 well)	In vitro
cCPE-Y306W/S313H ^b,f^ (10 μg/mL, in vitro) (5 ng/larva, in vivo) [200]	60% drop in TEER (in vitro) 4.3 increase in Rhod B-Dex flux (cerebellar central artery) (in vivo) 4.6 increase in Rhod B-Dex flux (middle mesencephalic central artery) (in vivo) 5.3 increase in Rhod B-Dex flux (metencephalic artery) (in vivo)	<3 h (in vitro) <1 h (in vivo)	48 h (in vitro) 3–4 h (in vivo)	Rhod B-Dex 10 kDa	Monoculture BBB: bEnd.3 Zebrafish Larvae (in vivo)	In vitroposterior cardinal vein injection (In vivo)
C1C2 ^b^ (200 μM) [202]	50% drop in TEER 8.25-fold increase in lucifer yellow 7-fold increase in AlexaFluor 680-dextran	<2 h	>24 h	Lucifer Yellow 444 Da AlexaFluor-680 3 kDa	Monoculture BBB: Primary mouse BMECs (24 well)	In vitro
D-aa-C5C2 ^b,g^ In vivo(3.5 μmol/kg) In vitro(300 μM) [208]	55% drop in TEER (in vitro-endo) 1.4-fold increase in Gd-DTPA (in vivo) 4-fold increase in Doxorubicin (in vitro—epithelial) 5.5-fold Lucifer Yellow flux (in vitro—epithelial) 3.75-fold increase in Fluorescein-Dex flux (in vitro—epithelial)	<4 h (in vivo) <12 h (in vitro)	4–12 h (in vivo) >48 h (in vitro)	Doxorubicin 544 Da Gd-DTPA 547 Da Lucifer Yellow 457 Da Fluorescein-Dex 10 kDa	Monoculture BBB: bEND.3 cells (12 well) Monoculture Epithelial: MDCKII cells (Cld-5 transfected) (12 well) Animal model: C57BL/6N mice, 10–19 weeks old (18–23 g)	In vitroTail vein injection (in vivo)
cCPE-Y306W/S313H ^b^ In vivo(360 nmol/kg) In vitro(120 μg/mL) [214]	97% drop in TEER 5.6-fold increase in ASO brain levels	<2 h (in vitro) <1 h (in vivo)	>120 h (in vitro) N.R (in vivo)	ASO (16 NCT’s) 5.3 kDa	Triple Culture BBB: Primary rat BCEC’s Primary rat Pericytes Primary rat astrocytes (24 well) Animal model: Wild-type female C57BL/6 mice (8–11 weeks)	Intravenous injection
Angubindin-1 ^b^ (10 mg/kg) [214]	90% drop in TEER (in vitro) 20-fold increase in ASO brain levels (in vivo)	<2 h (in vitro) <1 h (in vivo)	120 h (in vitro) <24 h (in vivo)	ASO (16 NCT’s) 5.3 kDa	Triple Culture BBB: Primary rat BCEC’s Primary rat Pericytes Primary rat astrocytes (24 well) Animal model: Wild-type female C57BL/6 mice (8–11 weeks)	Intravenous injection
Gintonin ^b^ (100 μg/mL, in vitro) (10 mg/kg, in vivo) [215]	~110-fold increase in Texas red-Dextran (in vitro) Approx. 4-fold increase in FITC Dextran Brain levels (in vivo) 41% increase in EPO levels in the CSF (in vivo)	<1 min (in vitro) <5 min (in vivo)	15–30 min (in vitro) >30 min (in vivo)	Texas red-Dextran (70 kDa) FITC-Dextran (10 kDa) EPO (34 kDa)	Monoculture BBB: HBMECs (24 well) Animal model: 8 Week Male Sprague Dawley rats (220–250 g)	Retro-orbital vein injection
M48 ^b^ (150 μg/mL) [217]	98% drop in TEER 3-fold increase in P(app) (Fluorescein) 3.5-fold increase in P(app) (Fluorescein-Dex)	<3 h	12–24 h	Fluorescein 376 Da Fluorescein-Dex 4 kDa	Triple Culture BBB: CMBMECs Rat cerebral astrocytes Rat cerebral pericytes (24 well)	In vitro
R9 ^b^ (150 μg/mL) [217]	95% drop in TEER 3.75-fold increase in P(app) (Fluorescein) 5.75-fold increase in P(app) (Fluorescein-Dex)	<3 h	12–24 h	Fluorescein 376 Da Fluorescein-Dex 4 kDa	Triple Culture BBB: CMBMECs Rat cerebral astrocytes Rat cerebral pericytes (24 well)	In vitro
Oligonucleotides						
Claudin-5 + Occludin siRNA ^b,h^ (10 pmol(each)/well) (20 μg(each)/mouse, 1 mg(each)/kg) [153]	2.8-fold increase in apical to basolateral permeability of FITC amyloid-β peptide (in vitro) 2.5-fold increase in apical to basolateral P_app_ of FITC amyloid-β peptide (in vitro) 17-fold increase in basolateral to apical permeability of FITC amyloid-β peptide (in vitro) 20-fold increase in basolateral to apical P_app_ of FITC amyloid-β peptide (in vitro) 2.4-fold increase in 3k biotin-dextran (in vivo)	<72 h (in vitro) (in vivo)	>74 h (in vitro) >72 h (in vivo)	Biotin-dextran 3 kDa	Monoculture BBB model: Bend.3 (24 well) Animal model: Tg2576 mice (~20 g)	Transwell luminal surface (in vitro) Transwell abluminal surface (in vitro) Tail vein injection (in vivo)
Claudin-5 siRNA ^b,h^ (10 pmol/well) (20 μg/mouse, 1 mg/kg) [153]	2.5-fold increase in apical to basolateral permeability of FITC amyloid-β peptide (in vitro) 2-fold increase in apical to basolateral P_app_ of FITC amyloid-β peptide (in vitro) 8.5-fold increase in basolateral to apical permeability of FITC amyloid-β peptide (in vitro) 8-fold increase in basolateral to apical P_app_ of FITC amyloid-β peptide (in vitro) 1.3-fold increase in 3k biotin-dextran (in vivo)	<72 h (in vitro) (in vivo)	>74 h (in vitro) >72 h (in vivo)	Biotin-dextran 3 kDa	Monoculture BBB model: Bend.3 (24 well) Animal model: Tg2576 mice (~20 g)	Transwell luminal surface (in vitro) Transwell abluminal surface (in vitro) Tail vein injection (in vivo)
Occludin siRNA ^b,h^ (10 pmol/well) (20 μg/mouse, 1 mg/kg) [153]	2.6-fold increase in apical to basolateral permeability of FITC amyloid-β peptide (in vitro) 2.3-fold increase in apical to basolateral P_app_ of FITC amyloid-β peptide (in vitro) 11-fold increase in basolateral to apical permeability of FITC amyloid-β peptide (in vitro) 10-fold increase in basolateral to apical P_app_ of FITC amyloid-β peptide (in vitro) No significant increase in 3k biotin-dextran (in vivo)	<72 h (in vitro) (in vivo)	>74 h (in vitro) >72 h (in vivo)	Biotin-dextran 3 kDa	Monoculture BBB model: Bend.3 (24 well) Animal model: Tg2576 mice (~20 g)	Transwell luminal surface (in vitro) Transwell abluminal surface (in vitro) Tail vein injection (in vivo)
Claudin-5 shRNA ^c,i^ (2 µL AAV sln) [223]	6.5-fold increase in Biotin (hippocampus) 3.6-fold increase in Biotin (mPFC)	<24 h	N.R	Biotin 600 Da	C57/BL6J mice (8–12 weeks)	Stereotaxic injection into hippocampus or mPFC
Claudin-5 siRNA ^b^ (20 μg/mouse) [224]	1.25-fold increase in Gd-DTPA	<24 h	3–7 days	Gd-DTPA 742 Da	C57/BL6 mice (20–30 g)	Tail vein injection
13-mer Toc-HDO ^c,i^ (50 mg/kg) [228]	55% reduction in efflux rate	<72 h	>72 h	^99m^Tc-ECD 436 Da	Wild-type C57BL/6 mice (7–10 weeks)	Intravenous injection

Abbreviations: ^99m^Tc-ECD (technetium (99mTc) bicistate), 13-mer Toc-HDO (13-mer α-tocopherol conjugated heteroduplex oligonucleotide), AFC (AlexaFluor 555–cadaverine), ASO (antisense oligonucleotide), BCECs (brain capillary endothelial cells), BMEC (brain microvascular endothelial cell), CF (carboxyfluorescein), Cld-5 (claudin-5), CMBMECs (cynomolgus monkey brain microvasculature endothelial cells), CSF (cerebrospinal fluid), Cy3-PD-hsiRNA (cyanine 3-labeled phosphocholine-docosahexanoic acid-hydrophobic siRNAs), EBA (Evans blue albumin), EBD (Evans blue dye), Endo (endothelial), Fluorescein-Dex (fluorescein isothiocyanate-labelled dextran), Gd-DTPA (gadolinium (III) diethylenetriaminepentaacetic acid), HRP (horseradish peroxidase), ICV (intracerebrovascular), mPFC (medial prefrontal cortex), NCT’s (nucleotides), N.R (not reported), NTU (no therapeutic tracer used), pPBMEC (porcine brain microvascular endothelial cells), RB 200 albumin (lissamine-rhodamine B200 albumin), RB 200 γ-globulin (lissamine-rhodamine B200 albumin and bovine γ-globulin), Rhod B-Dex (rhodamine B Dextran), sln (solution), TEER (trans endothelial electrical resistance). ^a^ Systemic distribution throughout the ipsilateral region. ^b^ Modulates the BBB paracellularly. ^c^ Modulates the BBB transcellularly. ^d^ Affinity for tumour tissue. ^e^ Pronounced tracer flux in the midbrain. ^f^ Pronounced tracer flux in the metencephalic artery. ^g^ Pronounced leakage in the periventricular region (in vivo). ^h^ Endothelial-selective. ^i^ Site-specific suppression of claudin-5 (HPC and mPFC). Note: For BBB in vitro models, agents were added to a transwell insert, unless otherwise specified.

#### 6.2.1. Future Prospects of BBB Modulation

##### Focused Ultrasound

Currently, over half (25) of the active clinical trials in the area of BBB modulation are considering the application of focused ultrasound-based BBB disruption (FUS BBBD) in the treatment of patients with CNS diseases (Figure 7). Additionally, the majority of those studies involve evaluation of this technology in non-combination studies (Table 5) aimed primarily at establishing safety and feasibility in the context of disease. FUS is still in the early stages of development as a platform technology (>136 clinical indications and 17 of 25 trials are still in the recruitment or pre-recruitment phase); however, early-phase FUS BBBD clinical trials have so far deemed the procedure to be safe [57,59] with one combination study showing early evidence of efficacy in treating gliomas [58].

Clinical trial progression in the area has been hindered by a number of challenges identified at the Focused Ultrasound & Blood–Brain Barrier Workshop held in 2017 by the Focused Ultrasound Foundation (FUF) [230]. For example, there is not yet an established pathway for drug/device combinations in the FUS field (i.e., FUS induced drug delivery) and it is difficult to see how regulatory agencies such as the FDA and EMA should ultimately assess and regulate it; as a drug or a device. The scientific literature reports wide instrumental parameter variation with MRgFUS BBBD, making it difficult to accurately compare studies. To provide further insight into this variation, the FUF have listed 19 key parameters that mediate FUS BBB opening [230]. It has been suggested by the foundation that FUS BBBD protocols might only be standardised by sub-classing protocols according to disease, device and drug type. In addition, the foundation highlighted the importance of standardising how to assess safety, BBB opening and drug delivery to the CNS. Providing standardised criteria will ultimately enable reliable evaluation that can support later phase success. The FUS clinical trials that are incorporating a drug-device combination (10) have all been initiated within the last 33 months except for one study, potentially indicating preliminary pathway development towards facilitating FUS drug-device evaluation.

A potential downside to the technology is the reliance of MRgFUS on MRI guidance which places a heavy dependency on expensive infrastructure currently not found in most clinics worldwide. If this technique is deemed safe and effective for treating a wide range of CNS diseases, patient demand will be immense and aggressive expansion of MRI infrastructure will be required as the current state would render MRgFUS BBBD an unfeasible approach.

Treatments that adopt single or infrequent dosing regimens such as gene therapy might be particularly suited to overcoming potential clinical limitations [55]. Recent advancement toward lowering the complex infrastructure required for FUS BBBD is the implantable medical device, SonoCloud^®^ developed by CarThera [231,232]. SonoCloud^®^ is a low intensity ultrasound transducer that is precisely implanted into the cranium of patients over the target site and removes the need for real-time MRI guidance, requiring only an ultrasound source. A drawback of the device requiring implantation following either surgical resection of a tumour, via burr hole or via routine surgery under local anaesthesia is also a strength of the device, as barriers to successfully delivering ultrasound via FUS such as skull thickness and unevenness are removed.

##### Small Molecules and Peptides

As mentioned prior, mannitol is indiscriminate with respect to BBB disruption; however, this may be advantageous in the context of therapy for CNS diseases affecting widespread brain tissue, such as neurological genetic disorders and metastasised cancers [233,234,235]. Indeed, it would seem that mannitol-induced BBBD may even complement MRgFUS, whereby FUS might treat more localised diseases. Noteworthy, the use of microcatheters designed to enable Superselective Intra-Arterial Cerebral Infusion (SIACI) has rejuvenated the focus on mannitol-based BBBD, with 11 clinical trials currently being undertaken in the area (Table 5). SIACI aims to overcome the issue of subtherapeutic dosing at tumour sites and the potential for focal toxicity resulting from an incomplete understanding of the relevant hydrodynamic and pharmacokinetic factors influencing drug administration at the BBB [79,236]. Interestingly, SIACI of mannitol, followed directly by infusion of the therapeutic agent (e.g., bevacizumab, cetuximab, or temozolomide) via the same catheter, has been shown to be safe and well tolerated [85,237]. Important also, is that this treatment modality compares favourably in terms of cost-effectiveness with standard I.V. dosing, potentially saving patients $1.3 million per year [79,238].

A recent preprint by Linville et al. describes the reversible opening of the BBB following administration of the 26-residue amphipathic peptide, melittin [239]. Melittin is a membrane active antimicrobial peptide that is the main pain-producing component (~50% wt.%) of bee venom [240,241] and has been reported to exhibit potent cytolytic and antimicrobial activities [242,243,244,245,246]. Intra-arterial injection of 3 μM melittin in mice resulted in localised cell contraction and subsequent leakage of Evan’s blue dye in the downstream ipsilateral hemisphere. Recent developments in peptide BBBD are lacking; however, significant advances have been made in the development of highly efficient peptide delivery vectors for cargoes such as nucleic acids [247,248].

##### Oligonucleotides

As of yet, gene silencing-mediated BBBD has only been tested in pre-clinical models. Oligonucleotides generally exhibit unfavourable pharmacokinetics in their unmodified form— in particular, siRNAs. Unmodified siRNA or ‘naked’ siRNA, when administered into the body, is subject to nuclease degradation and immune recognition within the bloodstream, resulting in rapid elimination and a short half-life [249]. To overcome this hurdle, conjugating a ligand that acts as a substrate for a receptor found exclusively or predominantly on the target cell, such as the brain microvascular endothelium (BMECs), has proven an effective approach. In addition to guiding siRNA to its target, the ligand promotes greater selectivity, reducing side effects and increasing its potency. Kuwahara et al. incorporated both locked nucleic acids and *α*-tocopherol as a guide-ligand to their ASO construct (Table 4), which silenced BMEC expression of P-glycoprotein by 55% [228]. Although the required dose was considerably high to achieve such efficacy (50 mg/kg), it nevertheless provided proof-of-concept for antisense oligonucleotide-based gene silencing of the BBB.

Nanoparticle encapsulation of siRNA can shield it from nucleases, prevent glomerular filtration and alter the surface charge to promote greater cell permeability [250]. In this regard, Keaney et al. [153] achieved highly efficacious and potent silencing of both claudin-5 and occludin in mice by complexing siRNA to the transfection agent in vivo-jetPEI™ by PolyPlus Transfection^®^. This cationic polymer complexes with siRNA to form nanoparticles of ~50 nm in size, preventing accumulation and subsequent glomerular filtration. Additionally, the cationic polyplex supports interactions with anionic syndecans on cell surfaces, stimulating endocytosis [251]. The endosomes containing the polyplex become acidified; however, the basic polyethyleneimine (PEI) consumes the influxing protons. The subsequent additional influx of protons and its counter ions triggers osmotic swelling, inducing rupturing and release of the polyplex into the cytoplasm [252]. The polyplex disassembles once in the cytoplasm, allowing the free siRNA to reach its binding partners and form the RISC complex [253]. The elegant multistep shuttling of siRNA by in vivo-jetPEI enables high affinity for brain microvascular endothelial cells upon intravenous administration. Despite in vivo-jetPEI being regarded as non-target specific without additional modifications made to the nanoformulation, the exceptionally anionic charge of BMECs relative to other endothelia may support its inherent affinity for this cell type [254]. This relatively simple nanoformulation provides an effective route for siRNA and potentially other therapeutics to reach BMECs efficiently upon intravenous administration.

Prolonged gene silencing is a caveat, however, that must be addressed comprehensively in future clinical trials when considering siRNA-mediated BBBD. While BBBD lasting minutes to an hour may not lead to extensive disruption of brain homeostasis, siRNA’s prolonged multi-day BBBD could potentially cause notable adverse effects. Addressing the effects of prolonged claudin-5 silencing, Campbell et al. developed an inducible short hairpin RNA against claudin-5 and incorporated it into an AAV vector [255]. Stereotaxic injection of this vector to the desired brain region provided a site-specific silencing of claudin-5 inducible upon doxycycline administration, that resulted in size-selective modulation of the BBB (~1 kDa limit). The lack of nuclear division in established endothelium enables long-term inducibility of claudin-5 silencing, facilitating the adoption of an infrequent dosing regimen—an important factor considering the highly invasive nature of the shRNA AAV administration [255]. Despite the invasiveness, this approach represents an excellent proof-of-concept towards the goal of achieving size-selective, inducible silencing of endothelial TJ proteins using siRNA.

##### A Combined Approach

The myriad of modulators reported in this review primarily increase BBB permeability paracellularly, circumventing the intricacies and limitations of transcellular transport. Despite this bypassing of the specialised BBB infrastructure, their presence remains a significant hinderance to CNS drug delivery. A recent study by de Gooijer et al. highlighted how paracellularly compromised (leaky) tumor vasculature in mice enabled increased levels of docetaxel to cross the brain tumor vasculature relative to normal BBB vasculature [256]. However, knockout of P-gp within this leaky BBB model further increased docetaxel levels in brain tumor tissue by up to 240%, indicating that paracellular BBB modulation does not necessarily ensure unimpeded access of drugs that possess affinity for efflux pumps. These findings highlight a potential limitation to paracellular BBB modulation, while instigating a call for the development of combination approaches that modulate the BBB both paracellularly and transcellularly, in a synchronised fashion.

##### BBB Modulation and Sterile Inflammation

Although completed phase I clinical trials (Table 6) have deemed MRgFUS to be safe and well tolerated within human populations [57,58,59], there has been a growing concern over recent years within this field about sterile inflammation imparted by the biophysical actions of FUS within animal models [257,258]. Initially reported by Jordão et al. [259] in 2013, Kovacs et al. supported this finding using proteomic and transcriptomic analysis to reveal the immediate generation of a damage-associated molecular pattern (DAMP) response upon acoustic cavitation induced by FUS. Additional histological analysis revealed macrophage infiltration within the parenchyma, indicative of sterile inflammation and an innate immune response [260]. Inflammation was temporary in this report, lasting up to 24 h; however, a follow-up study using a large microbubble dose, in addition to a high-pressure amplitude, triggered elevated immunoreactivity lasting 7 weeks coupled with glial scar formation [261]. FUS-induced inflammation can be optimised to limit the severity of parenchymal sterile inflammation [262,263,264]; however, the safety of even these conditions must be thoroughly assessed, with particular attention paid to the long-term effects of repeated FUS sessions.

Importantly, FUS is not the only approach to report this occurrence. Mannitol-based BBB modulation was reported to induce a sterile inflammation response (SIR) resemblant to FUS just minutes after intracarotid arterial administration of mannitol [265]. Burks et al. optimistically regard this finding as a potential application for neuroimmunomodulation, and while it could indeed have uses in various immune-based therapies for CNS diseases such as CAR-T in treating glioblastoma [266], the systemic effect of mannitol-based BBB disruption upon the CNS demands careful and diligent monitoring of this phenomenon in future clinical trials.

While a large cohort of studies deem the reported SIR to be safe and manageable, inducing immune cell trafficking across the BBB may not be compatible in treating autoimmune CNS diseases such as multiple sclerosis; these modulative approaches may aggravate an already highly active disease. Hence, alternative approaches to BBB modulation will likely need to be developed alongside these approaches to increase the likelihood of delivering successful therapies for general and autoimmune CNS diseases alike.

**Table 4 pharmaceutics-13-01980-t004:** Chemical Properties of BBB modulators.

**Small Molecule**	**Chemical Formula**	**Molecular Mass (Da)**	**Chemical Structure**
Hyperosmolar Agents
Mannitol	C_6_H_14_O_6_	182.17	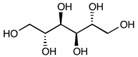
Arabinose	C_5_H_10_O_5_	150.13	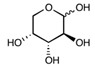
Inflammatory Mediators
Histamine	C_5_H_9_N_3_	111.15	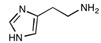
Leukotriene C4	C_30_H_47_N_3_O_9_S	625.77	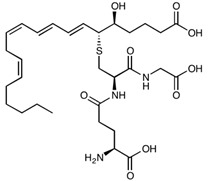
Leukotriene D4	C_25_H_40_N_2_O_6_S	496.66	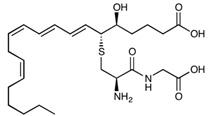
Leukotriene E4	C_23_H_37_NO_5_S	439.61	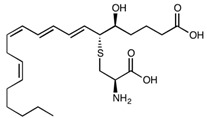
Alkylglycerols
1-O-Pentylglycerol	C_8_H_18_O_3_	162.23	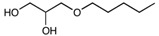
2-O-hexyldiglycerol	C_12_H_26_O_5_	250.33	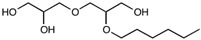
Other
Sodium Caprate	C_10_H_19_NaO_2_	194.25	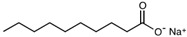
Regadenoson	C_15_H_18_N_8_O_5_	390.35	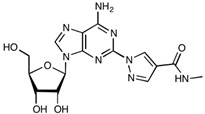
Fingolimod	C_19_H_33_NO_2_	307.47	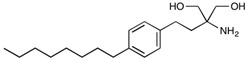
NIBR-0213	C_27_H_29_ClN_2_O_3_	464.98	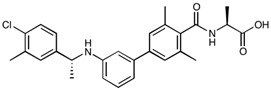
NS1619	C_15_H_8_F_6_N_2_O_2_	362.23	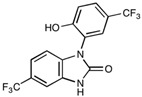
NEO100	C_10_H_16_O	152.23	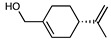
M01	C_30_H_28_N_4_O_5_	524.57	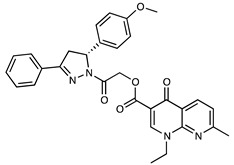
**Peptide/Peptidomimetic**	**Peptide Sequence**	**Molecular Mass (Da)**	**Chemical Structure**
ADT-6	Ac-ADTPPV-NH2	639.70	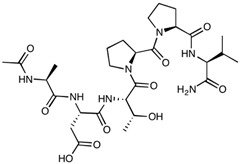
ADTC5	Cyclo(1,7)Ac-CDTPPVC-NH2	772.89	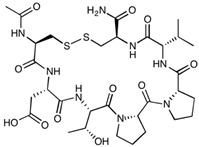
AT-1002	FCIGRL	707.88	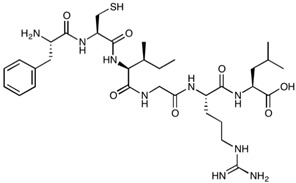
AT-1002 (Allyl-Gly)	H-Phe-(Allyl)Gly-Ile-Gly-Arg-Leu-OH	701.86	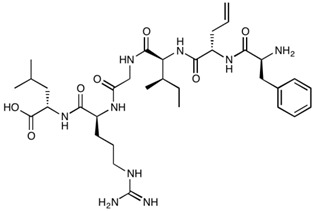
Bradykinin	RPPGFSPFR	1060.21	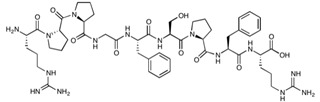
cCPE	SSYSGNYPYSILFQKF	1901.08	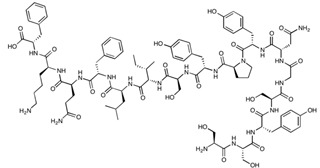
cCPE-Y306W/S313H	SHYSGNYPWSILFQKF	1974.18	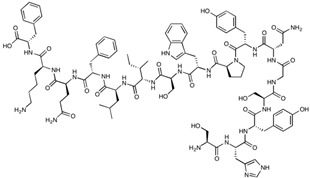
C1C2	SSVSQSTGQIQSKVDSLLNLNSTQATR-NH2	2835.05	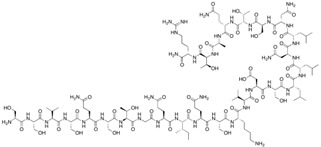
cHAVc3	Cyclo(1,6)Ac-CSHAVC-NH2	657.76	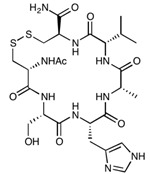
HAV-6	Ac-SHAVSS-NH2	627.65	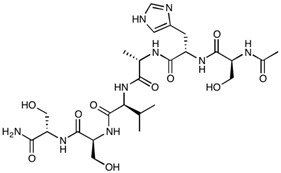
PN-159	KLALKLALKALKAALKLA-NH2	1876.46	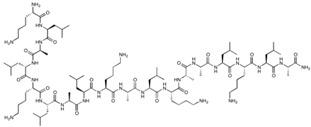
RMP-7	H-Arg-Pro-Hyp-Gly-2Thi-Ser-Pro-βTyr(Me)-Arg-OH	1098.28	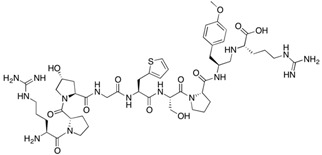
7-mer (PN78)	FDFWITP	925.04	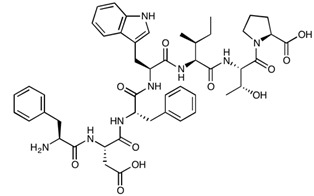
**Protein**	**Molecular Mass (Da)**	**Amino Acid Sequence**
Zonula occluden toxin	44,903.41	
∆G	12,852.67	
Angubindin-1	27,020.19	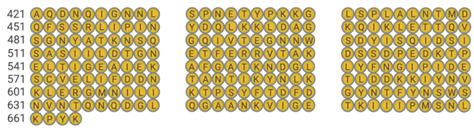
**Oligonucleotide**	**Molecular Mass (Da)**	**Oligonucleotide Sequence**
Claudin-5 siRNA	13,352.40	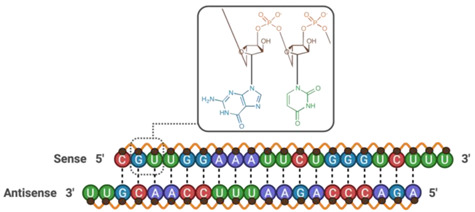
Occludin siRNA	13,307.37	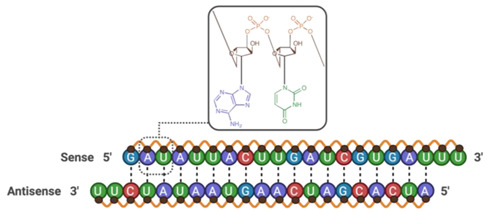
13-mer Toc-HDO	8984.37	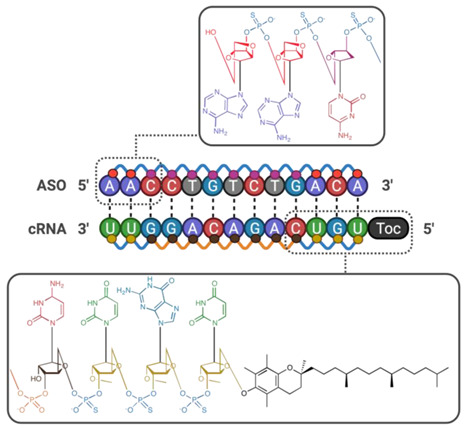

**Small molecule:** Small molecule BBB disruptors have been researched for over five decades and are categorised into three main families; hyperosmolar agents, alkylglycerols and inflammatory mediators. Newer BBB disruptors have steered away from these conventional categories (Other), with the majority showing efficacy while avoiding intracarotid injection as a route of administration. **Peptidic:** From phage display to structural simplification of protein-based toxins, various classes of peptides disrupt the blood–brain barrier in numerous ways. This diversity renders peptide-based BBBD a flourishing area, with promising pre-clinical evidence currently being published. Letter abbreviations of peptide sequences correspond to the amino acid alphabet. **Protein:** Zonula toxin (ZOT) is a 45 kDa protein that sheds off a 12 kDa active component called ∆G. Further isolation of the active component of ∆G was found to be residues 288–293, which is known as AT-1002. Although AT-1002 retains ∆G’s efficacy in BBB modulation, its potency is reduced 5-fold, with dependency on supplemental protease inhibitors to prevent its degradation in the digestive tract. Angubindin-1, a 27 kDa protein derived from a section of the Ib domain of *Clostridium perfringens* iota-toxin (amino acids 421–664) has been shown to disrupt tricellular TJs (tTJs) through binding to angulin-1 and -3—two components that each in conjunction with tricellulin form tTJs [214]. In addition to its BBB-disrupting abilities, its effect on epithelial tissue is limited, providing a dimension of target selectivity. Letter abbreviations of peptide sequences correspond to the amino acid alphabet. **Oligonucleotides:** Gene silencing of tight junction proteins and efflux pumps through silencing RNA (siRNA) and antisense oligonucleotides (ASOs) possess unrivalled target precision. Despite their problematic pharmacokinetics in ‘naked’ form, chemical modification provides a solution. 
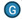
 Guanine, 
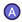
 Adenine, 
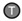
 Thymine, 
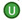
 Uracil, 
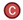
 Cytosine, 
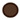
 Ribose sugar unit, 
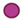
 Deoxyribose Sugar unit, 
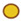
 2′-O-methyl ribose, 
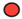
 Locked deoxyribose, 
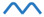
 Phosphothioate bonds, 
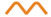
 Phosphonate bonds, 
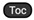
 α-Tocopherol.

**Table 5 pharmaceutics-13-01980-t005:** Active NIH Clinical Trials in Therapeutic Blood–Brain Barrier Disruption (BBBD).

Title	Condition(s)	BBBD Method	Phase	Enrolment	Start Date	Status
Focused Ultrasound (Non-combination studies)						
Assessment of Initial Efficacy and Safety of High Intensity Focused Ultrasound ‘ExAblate 4000 Type 2’ for Blood-Brain Barrier Disruption in Patients With Alzheimer’s Disease	Alzheimer’s Disease	Exablate	N.A.	6	10 April 2020	Active, not recruiting
Blood-Brain Barrier Opening in Alzheimer’ Disease (BOREAL1)	Alzheimer’s Disease	SonoCloud	I/II	10	26 June 2017	Active, not recruiting
Blood-Brain Barrier Opening Using MR-Guided Focused Ultrasound in Patients With Amyotrophic Lateral Sclerosis	Amyotrophic Lateral Sclerosis	MRgFUS	N.A.	8	13 April 2018	Active, not recruiting
Non-invasive Blood-brain Barrier Opening in Alzheimer’s Disease Patients Using Focused Ultrasound	Alzheimer’s Disease	MRgFUS	N.A.	6	1 October 2020	Recruiting
A Study to Evaluate Temporary Blood-Brain Barrier Disruption in Patients With Parkinson’s Disease Dementia	Parkinson’s Disease Dementia	MRgFUS	N.A.	10	26 November 2018	Active, not recruiting
The Use of Focused Ultrasound and Microbubble Infusion for Altering Brain Perfusion and the Blood-Brain Barrier	Low Grade Glioma	MRgFUS	N.A.	15	1 February 2020	Not yet recruiting
Assessment of Safety and Feasibility of ExAblate Blood-Brain Barrier (BBB) Disruption for Treatment of Glioma	Glioblastoma	ExAblate	N.A.	20	16 October 2018	Recruiting
Assessment of Safety and Feasibility of ExAblate Blood-Brain Barrier (BBB) Disruption	Glioma	ExAblate	N.A.	20	26 March 2019	Recruiting
ExAblate Blood-Brain Barrier Opening for Treatment of Alzheimer’s Disease	Alzheimer’s Disease	ExAblate	N.A.	30	6 December 2018	Recruiting
ExAblate Blood-Brain Barrier (BBB) Disruption for the Treatment of Alzheimer’s Disease	Alzheimer’s Disease	ExAblate	N.A.	20	28 September 2018	Recruiting
ExAblate Blood-Brain Barrier Disruption (BBBD) for Planned Surgery in Suspected Infiltrating Glioma	Glioma	ExAblate	N.A.	15	18 October 2018	Active, not recruiting
ExAblate Blood-Brain Barrier Disruption for Glioblastoma in Patients Undergoing Standard Chemotherapy	Glioblastoma multiforme	ExAblate	N.A.	10	28 August 2018	Recruiting
Blood-Brain Barrier Disruption (BBBD) Using MRgFUS in the Treatment of Her2-positive Breast Cancer Brain Metastases	Breast cancer Brain metastases	ExAblate	N.A.	10	18 September 2019	Recruiting
Safety and Effectiveness of Blood-Brain Barrier Disruption (BBBD) in Subjects With Suspected Infiltrating Glioma (BBBD)	Glioma	ExAblate	N.A.	120	1 December 2021	Not yet recruiting
Assessment of Safety and Feasibility of ExAblate Blood-Brain Barrier (BBB) Disruption in GBM Patients	Glioma	ExAblate	N.A.	5	15 September 2021	Not yet recruiting
Focused Ultrasound (Drug-device Combination studies)						
Exablate Blood-Brain Barrier Disruption With Carboplatin for the Treatment of rGBM	Glioblastoma	Exablate	I/II	40	13 October 2020	Recruiting
Ultrasound-based Blood-brain Barrier Opening and Albumin-bound Paclitaxel for Recurrent Glioblastoma (SC9/ABX)	Glioblastoma	SonoCloud-9	I/II	37	August 2020	Recruiting
Blood-Brain-Barrier Disruption With Cerezyme in Patient’s With Parkinson’s Disease Dementia	Parkinson disease dementia	ExAblate	N.A.	6	16 July 2020	Active, not recruiting
Blood-Brain Barrier Disruption Using Transcranial MRI-Guided Focused Ultrasound	Brain Tumor	ExAblate	N.A.	10	October 2014	Active, not recruiting
Exablate Blood-Brain Barrier Disruption for the Treatment of rGBM in Subjects Undergoing Carboplatin Monotherapy	Glioblastoma	ExAblate	I/II	30	6 November 2020	Recruiting
Safety and Efficacy of Transient Opening of the Blood-brain Barrier (BBB) With the SonoCloud-9	Adult glioblastoma	SonoCloud-9	I/IIa	30	18 February 2019	Active, not recruiting
Safety and Efficacy of Sonocloud Device Combined With Nivolumab in Brain Metastases From Patients With Melanoma	Metastatic melanoma	SonoCloud	I/II	21	24 October 2019	Recruiting
Efficacy and Safety of NaviFUS System add-on Bevacizumab (BEV) in Recurrent GBM Patients	Glioblastoma	NaviFUS system	N.A.	10	21 July 2020	Recruiting
Non-Invasive Focused Ultrasound (FUS) With Oral Panobinostat in Children With Progressive Diffuse Midline Glioma (DMG)	Diffuse midline glioma	FUS	I	15	July 2021	Recruiting
Innovative SonoCloud-9 Device for Blood Brain Barrier Opening in First Line Temozolomide Glioblastoma Patients. (SonoFIRST)	Glioblastoma	SonoCloud-9	II	66	11 September 2021	Recruiting
Laser Heat Ablation						
Using MRI-Guided Laser Heat Ablation to Induce Disruption of the Peritumoral Blood-Brain Barrier to Enhance Delivery and Efficacy of Treatment of Pediatric Brain Tumors	Glioma	MRI-guided laser heat ablation	II	12	14 August 2015	Recruiting
MK-3475 in Combination With MRI-guided Laser Ablation in Recurrent Malignant Gliomas	Malignant glioma	MRI-guided laser heat ablation	I/II	58	5 August 2015	Active, not recruiting
Surgery						
Surgical Tissue Flap to Bypass the Blood-Brain Barrier in GBM	Glioblastoma multiforme	Temporoparietal fascial or Pericranial surgical flap	N.A.	10	27 July 2018	Recruiting
Laparoscopically Harvested Omental Free Tissue Autograft to Bypass the Blood-Brain Barrier (BBB) in Human Recurrent Glioblastoma Multiforme (rGBM)	Glioma	Laparoscopically harvested omental free flap	I	10	6 January 2020	Recruiting
Small Molecule						
Determining Dose of Regadenoson Most Likely to Transiently Alter the Integrity of the Blood-Brain Barrier in Patients With High Grade Gliomas	High grade glioma	Regadenoson	I	45	6 December 2019	Recruiting
Melphalan, Carboplatin, Mannitol, and Sodium Thiosulfate in Treating Patients With Recurrent or Progressive CNS Embryonal or Germ Cell Tumors	CNS tumours	Mannitol	I/II	55	9 July 2009	Active, not recruiting
Carboplatin, Melphalan, Etoposide Phosphate, Mannitol, and Sodium Thiosulfate in Treating Patients With Previously Treated Brain Tumors	Glioma	Mannitol	I/II	43	15 September 2005	Recruiting
Methotrexate, Mannitol, Rituximab, and Carboplatin in Treating Patients With Newly Diagnosed Primary Central Nervous System Lymphoma	CNS lymphoma	Mannitol	I/II	81	14 October 2005	Recruiting
Super-selective Intra-arterial Repeated Infusion of Cetuximab for the Treatment of Newly Diagnosed Glioblastoma	Glioblastoma	Mannitol (SIACI)	I/II	33	June 16	Recruiting
Super-selective Intra-arterial Cerebral Infusion of Trastuzumab for the Treatment of Cerebral Metastases of HER2/Neu Positive Breast Cancer	Neoplasm metastasis	Mannitol (SIACI)	I	48	August-2021	Recruiting
Super-Selective Intraarterial Cerebral Infusion Of Temozolomide (Temodar) For Treatment Of Newly Diagnosed GBM And AA	Glioma	Mannitol (SIACI)	I	30	August 2010	Active, not recruiting
Repeated Super-selective Intraarterial Cerebral Infusion Of Bevacizumab Plus Carboplatin For Treatment Of Relapsed/Refractory GBM And Anaplastic Astrocytoma	Glioma	Mannitol (SIACI)	I/II	54	September 2011	Suspended
Repeated Super-selective Intraarterial Cerebral Infusion of Bevacizumab (Avastin) for Treatment of Relapsed GBM and AA	Glioma	Mannitol (SIACI)	I/II	54	October 2010	Recruiting
Repeated Super-Selective Intraarterial Cerebral Infusion of Bevacizumab (Avastin) for Treatment of Newly Diagnosed GBM	Glioblastoma multiforme	Mannitol (SIACI)	I/II	25	February 2013	Recruiting
Intraarterial Infusion Of Erbitux and Bevacizumab For Relapsed/Refractory Intracranial Glioma In Patients Under 22	Glioma	Mannitol (SIACI)	I/II	30	June 2013	Recruiting
Super Selective Intra-arterial Repeated Infusion of Cetuximab (Erbitux) With Reirradiation for Treatment of Relapsed/Refractory GBM, AA, and AOA	Glioma	Mannitol (SIACI)	II	37	May 2016	Recruiting
IA Carboplatin + Radiotherapy in Relapsing GBM	Glioblastoma multiforme	Intra-arterial chemotherapy	II	35	10 July 2018	Recruiting
Miscellaneous						
TMS Electrochemotherapy for Glioblastoma Multiforme	Glioblastoma	TMS	II	20	January 2015	Suspended
The Danish Neuropsychological Study on the Adverse Effects of ECT	Depressive disorder	Electroconvulsive therapy	N.A.	290	12 November 2020	Recruiting
CED of MTX110 Newly Diagnosed Diffuse Midline Gliomas	Gliomas	Convection enhanced delivery	I	9	10 March2020	Recruiting

Abbreviations: TMS (Transcranial magnetic stimulation). Data from ClinicalTrials.gov (accessed on 19 October 2021).

## 7. Conclusions

Gaining control over BBB regulation opens up the possibility of facilitating delivery of targeted therapeutics to treat brain tumours (e.g., glioblastoma) and neurodegenerative disorders. This review has highlighted the key developments helping to drive the field forward and enable both the potent and selective modulation of the BBB. Due to the extensive surface area of the vascular system, a systemically distributed compound that avoids a device or surgical conjunctive therapy will likely need to be of exceptional potency to facilitate efficacious BBB opening and negate untoward side effects. In this regard, we strongly advocate investigation into additional targets and pathways that may offer gains in selectivity over other areas of the body. The latest clinical trial data points towards the potential future implementation of a drug–device combination using either FUS or SIACI to ensure highly localised drug delivery to the CNS, and stands to revolutionise current treatment options once the safety profiles of these approaches have been comprehensively investigated.

**Table 6 pharmaceutics-13-01980-t006:** Completed NIH Clinical Trials in Therapeutic blood–Brain Barrier Disruption (BBBD).

Trial	Condition(s)	BBBD Method	Phase	Enrolment	Start Date	End Date
Ultrasound						
Evaluation of Blood-Brain Barrier Integrity and Structural Abnormalities in MPS IIIB Patients Using Multimodal Magnetic Resonance Imaging	MPS IIIB (Sanfilippo B Syndrome)	MRgFUS	N.A	5	December 2013	May 2014
Blood-Brain-Barrier Opening Using Focused Ultrasound With IV Contrast Agents in Patients With Early Alzheimer’s Disease	Alzheimer’s Disease	ExAblate	I	6	December 2016	December 2017
Safety of BBB Opening With the SonoCloud	Glioma	SonoCloud	I/II	27	July 2014	July 2018
Safety of BBB Disruption Using NaviFUS System in Recurrent Glioblastoma Multiforme (GBM) Patients	Glioblastoma multiforme	NaviFUS system	N.A	6	August 2018	June 2019
Laser Heat Ablation						
MRI-Guided Laser Surgery and Doxorubicin Hydrochloride in Treating Patients With Recurrent Glioblastoma Multiforme	Glioblastoma	MRI-guided laser heat ablation	I/IIa	37	August 2013	May 2018
Small Molecule						
Methotrexate, Cyclophosphamide, and Etoposide Phosphate Given With Osmotic Blood-Brain Barrier Disruption Plus Dexamethasone and Cytarabine in Treating Patients With Primary CNS Lymphoma	Lymphoma	Osmotic BBBD (unspecified agent)	II	22	January 2000	July 2006
Brain Interstitium Temozolomide Concentration Pre and Post Regadenoson Administration	Blood–brain barrier defect	Regadenoson	I	6	February 2015	February 2018
Super-Selective Intraarterial Cerebral Infusion of Cetuximab (Erbitux) for Treatment of Relapsed/Refractory GBM and AA	Glioma	Mannitol (SIACI)	I	15	December 2009	January 2016
Super-Selective Intraarterial Intracranial Infusion of Avastin (Bevacizumab)	Glioma	Mannitol (SIACI)	I	30	July 2009	January 2014
Low-dose Intra-arterial Bevacizumab for Edema and Radiation Necrosis Therapeutic Intervention (LIBERTI)	Radiation Necrosis	Mannitol (IA)	II	10	November 2016	June 2019
Peptides						
A Pediatric Phase I Trial of RMP-7 and Carboplatin in Brain Tumors	Gliomas	RMP-7	I	30	April 1996	March 2000
The Safety and Effectiveness of RMP-7 Plus Amphotericin B in Patients With HIV and Cryptococcal Meningitis	Viral/Fungal Infections	RMP-7	I	N.R	August 2001 *	June2005 *
Lobradimil and Carboplatin in Treating Children With Brain Tumors	Brain and CNS Tumors	RMP-7	II	146 (max) *	March 1998	April 2003
Radiation Therapy Plus Carboplatin and Lobradimil in Treating Children With Newly Diagnosed Brain Stem Gliomas	Brain and CNS Tumors	RMP-7	I	13	February 2001	September 2005
Transcranial Magnetic Stimulation						
Effect of Deep TMS on the Permeability of the BBB in Patients With Glioblastoma Multiforme: a Pilot Study	Glioblastoma multiforme of the brain	dTMS	II	15	November 2014	May 2015
Electroconvulsive Therapy						
Exploring Effects of Electroconvulsive Therapy on the Human Brain in Depression—The Danish ECT/MRI Study	Major depressive disorder	Electroconvulsive therapy	N.A	60	August 2017	June 2020

Abbreviations: SIACI (superselective intra-arterial cerebral infusion), dTMS (deep transcranial magnetic stimulation). * Estimated from limited information provided by the NIH database. Data from ClinicalTrials.gov (accessed on 19 October 2021).

## Figures and Tables

**Figure 1 pharmaceutics-13-01980-f001:**
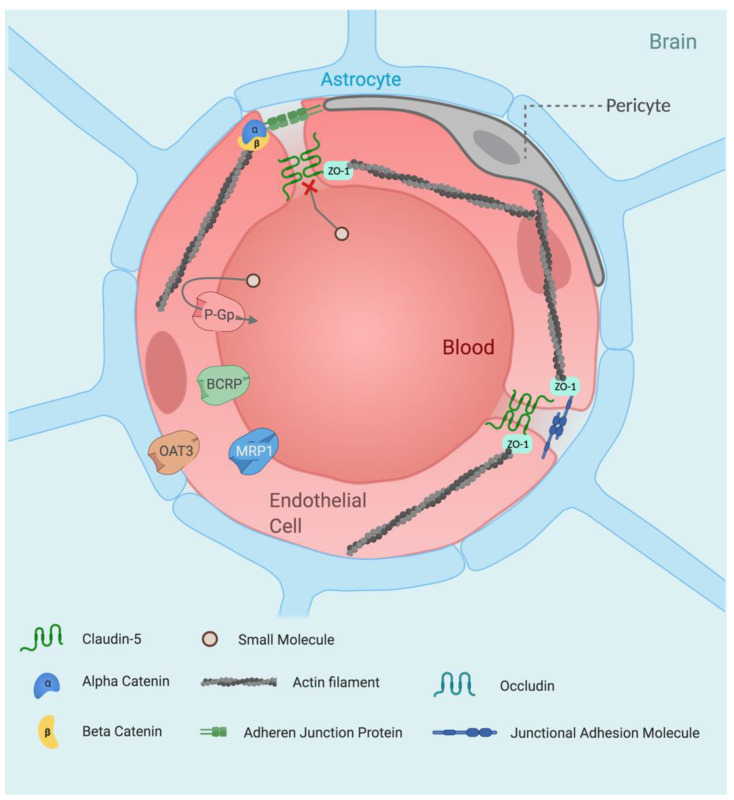
The healthy blood–brain barrier. The blood–brain barrier is comprised primarily of brain microvascular endothelial cells (BMECs) which create a physical barrier, arising through the formation of tight junctions at endothelial cell interfaces by claudins, occludin and junctional adhesion molecules. The formation of tight junctions restricts paracellular flux greatly while high expression of efflux proteins on the endothelial luminal surface hinders passive transcellular diffusion. In addition to BMECs, the BBB is comprised of astrocytes and pericytes. These cells provide less structural support with regard to BBB formation and function primarily in barrier regulation. Pericytes possess a broad range of functions in vascular regulation [15], while astrocytes provide a cellular link between the nervous tissue and the vascular system [16,17]. The organisation of these cell types forms the neurovascular unit, which is the building block that makes up the BBB. Abbreviations: BCRP (breast cancer resistance protein), OAT3 (organic anion transporter 3), P-Gp (P-glycoprotein), ZO-1 (zonula occludens-1).

**Figure 2 pharmaceutics-13-01980-f002:**
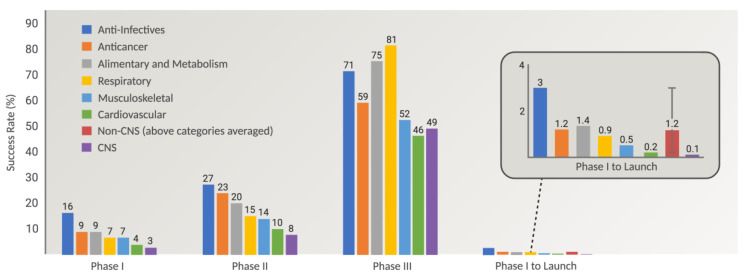
Success rates of clinical trial phases categorised by disease type from 2010–2017. The CMR R&D Performance Metrics reported the average clinical phase success rate of pharmaceutical drugs that treat the most common disease areas from 2010–2017. Combining the success rate of each phase provides the success rate probability of a drug from phase I to launch. Averaging the combined phase success rates of non-CNS drug categories leads to an average 12-fold greater success rate relative to CNS drugs, highlighting the high failure rate of CNS drug candidates in drug development. Data taken from ref. [19], with additional extrapolations made.

**Figure 3 pharmaceutics-13-01980-f003:**
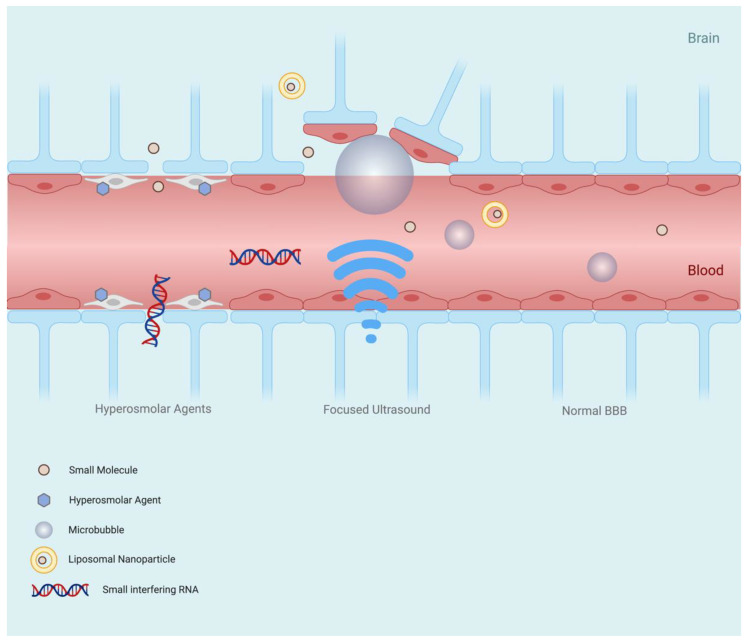
Blood–brain barrier modulation mechanism of focused ultrasound and hyperosmolar agents. Hyperosmolar agents disrupt the blood–brain barrier by dehydrating endothelial cells in the vascular lumen, causing them to contract. The shrinking of the cells leads to the opening of TJ gaps, enabling enhanced and indiscriminate paracellular flow of solutes. Exposure of microbubbles within brain capillaries to ultrasound waves induces rapid expansion and contraction, disrupting the microvascular endothelial cells of the blood–brain barrier and creating cavitation sites at a sensitivity greater than direct FUS. Additionally, the power required to induce blood–brain barrier disruption is lowered by 100-fold, falling below the levels required to damage tissue.

**Figure 4 pharmaceutics-13-01980-f004:**
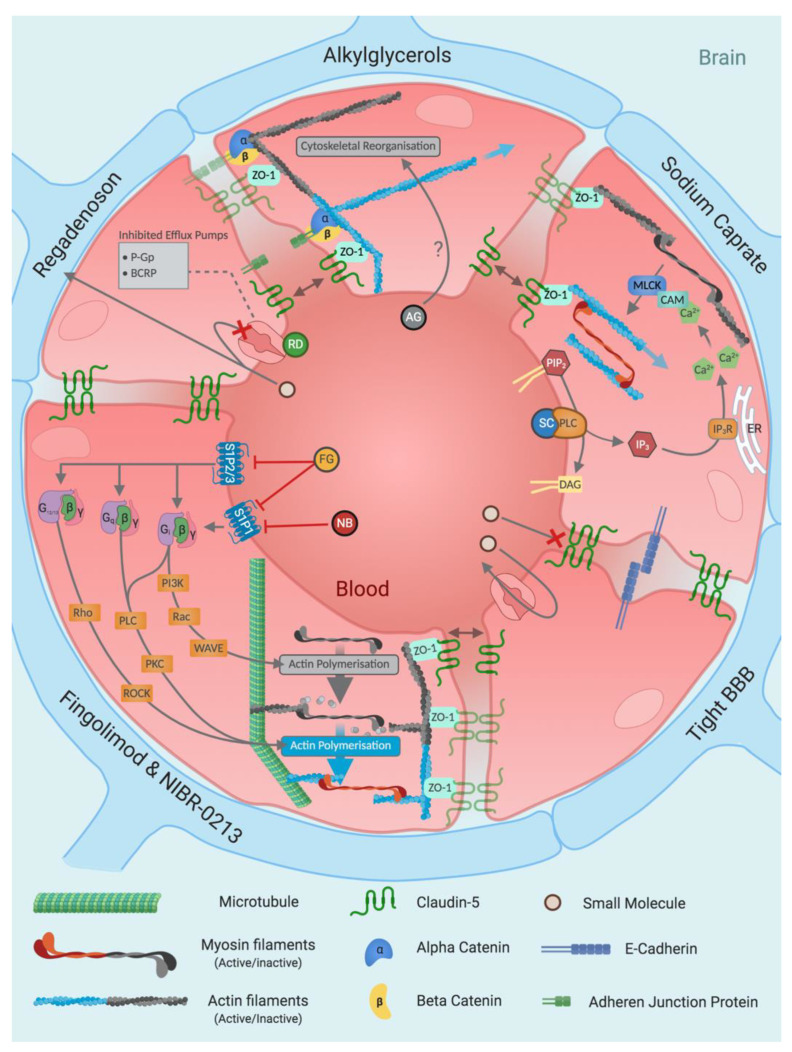
Mechanism summary for blood–brain barrier-modulating small molecules. Small molecules are a diverse category of blood–brain barrier modulators that primarily induce paracellular flux by altering tight junctions via indirect interactions through rearrangement of the cytoskeleton. Abbreviations: AG (alkylglycerol), BCRP (breast cancer resistant protein), β (beta GPCR subunit), *γ* (gamma GPCR subunit), CAM (calmodulin), DAG (diglyceride), ER (endoplasmic reticulum), FG (fingolimod), G_12/13_ (G12/G13 alpha GPCR subunits), G_i_ (Gi alpha GPCR subunit), G_q_ (Gq alpha GPCR subunit), IP_3_ (inositol triphosphate), IP_3_R (inositol triphosphate receptor), MLCK (myosin light chain kinase), NB (NIBR-0213), P-GP (P-glycoprotein), PI3K (phosphoinositide 3-kinase), PIP_2_ (phosphatidylinositol 4,5-bisphosphate), PKC (protein kinase C), PLC (phospholipase C), RD (regadenoson), ROCK (Rho-associated protein kinase), SC (sodium caprate), WAVE (WASP-family verprolin homologous protein), ZO-1 (zonula occludens-1).

**Figure 5 pharmaceutics-13-01980-f005:**
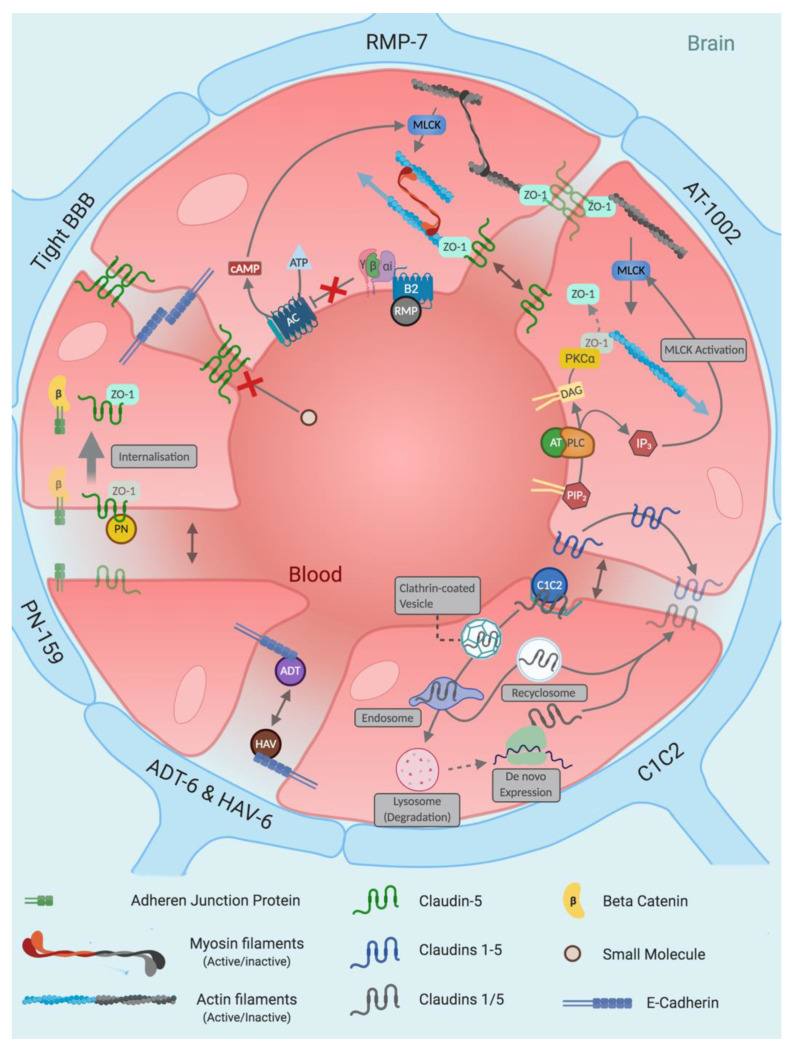
Mechanism summary for blood–brain barrier-modulating peptides. Similar to small molecule blood–brain barrier modulators, peptides affect tight junction integrity in a diverse manner, either through direct interruption of TJ protein trans interactions or by altering an indirect downstream effector. Abbreviations: *α*i (Gi alpha GPCR subunit), AC (adenylate cyclase), ADT (ADT-6 or ADTC5), AT (AT-1002), ATP (adenosine triphosphate), β (beta GPCR subunit), B2 (bradykinin receptor 2), *γ* (gamma GPCR subunit), cAMP (cyclic adenosine monophosphate), DAG (diglyceride), HAV (HAV-6 or cHAVc3), IP_3_ (inositol triphosphate), MLCK (myosin light chain kinase), PIP_2_ (phosphatidylinositol 4,5-bisphosphate), PLC (phospholipase C), PN (PN-159), RMP (RMP-7).

**Figure 6 pharmaceutics-13-01980-f006:**
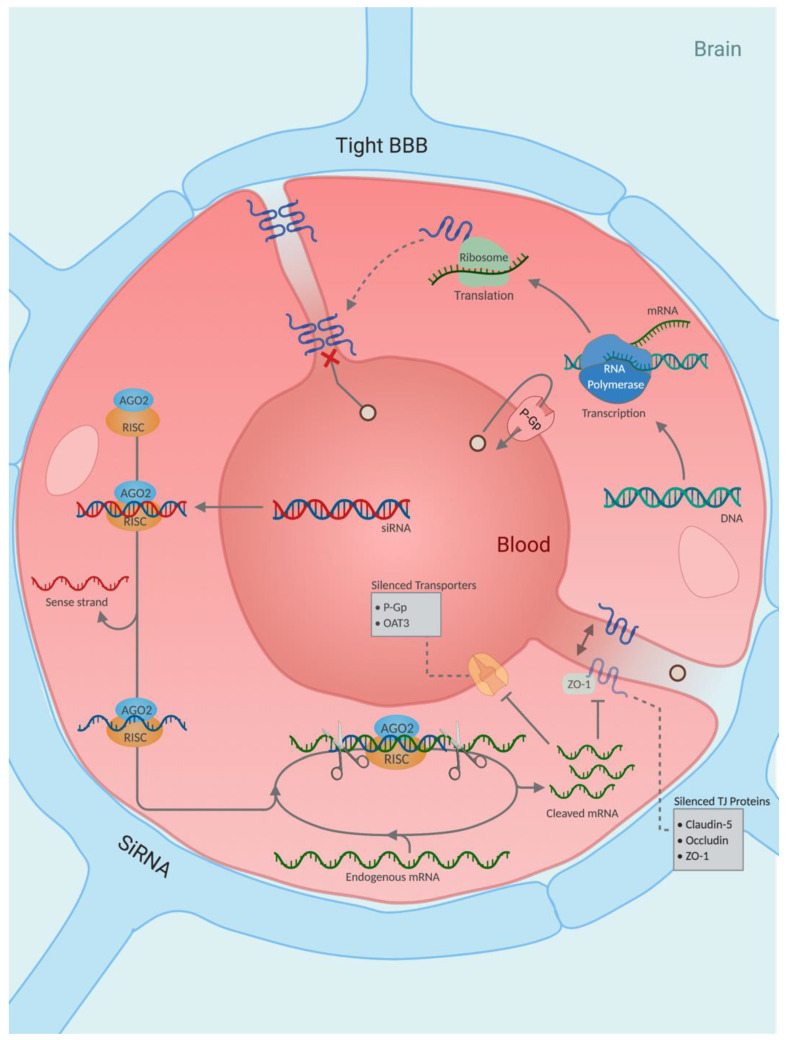
SiRNA Mechanism for blood–brain barrier modulation. siRNAs are derived from double-stranded RNAs of approximately 21 base pairs in length, that when introduced into a cell induce mRNA degradation for RNA of complementary nucleotide sequences, preventing the translation of a particular gene. siRNA has been used to silence transporters and tight junction proteins to modulate the blood–brain barrier. Abbreviations: AGO2 (protein argonaute-2), OAT3 (organic anion transporter 3), P-Gp (P-glycoprotein), RISC (RNA-induced silencing complex), ZO-1 (zonula occludens1).

**Figure 7 pharmaceutics-13-01980-f007:**
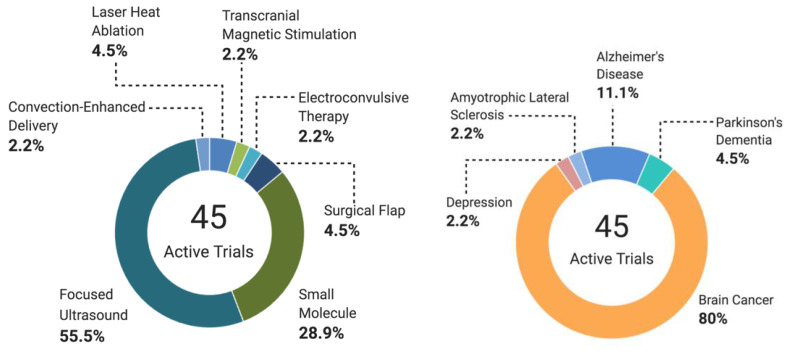
Summary of active BBBD clinical trials categorised by modulation method (**left**) and associated disease investigated (**right**). Currently 45 active clinical trials involving Blood–brain barrier Disruption are taking place. The majority of these trials are using focused ultrasound technology to facilitate greater efficacy of anticancer drugs to treat various gliomas.

## Data Availability

Not applicable.

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
