# Peer review of "Modulating the Blood–Brain Barrier: A Comprehensive Review"

_pharmaceutics, 2021, doi:10.3390/pharmaceutics13111980_

Round 1

Reviewer 1 Report

This is a very informative and comprehensive review. Nevertheless , I have some comments

1) in figure 1 the Mrp proteins should be included. 

2) Modulation of the BBB by inhibition of ABC transporters is only casually mentioned. However, there are many studies showing that such inhibition might have significant impact on drug uptake. This should also be included in the manuscript. 

Author Response

This is a very informative and comprehensive review. Nevertheless , I have some comments

  • In figure 1 the Mrp proteins should be included. 

Included an MRP efflux transporter in Fig 1 (MRP1)

  • Modulation of the BBB by inhibition of ABC transporters is only casually mentioned. However, there are many studies showing that such inhibition might have significant impact on drug uptake. This should also be included in the manuscript.

We realise that although we included BBB modulators that inhibit ABC transporters within the review:

  • Regadenoson inhibits P-gp (ABCB1) and BCRP (ABCG2) leading to increased BBB permeability
  • RNA silencing of P-gp increases drug BBB permeability

Reviewer 1 noted that we may have underrepresented this target of BBB modulation. To furtherly promote the importance of ABC transporters in BBB modulation, we incorporated this key paper by de Gooijer et al  (https://pubmed.ncbi.nlm.nih.gov/33521698/) that highlights the significant influence of P-gp in drug delivery to the CNS even when the BBB is opened paracellularly. By incorporating this paper within the review (Pg 33 A Combined Approach Section), we discuss how efflux transporter inhibition in combination with paracellular modulation is an important consideration to optimise the efficacy of BBB modulation hence drug delivery to the brain.

Reviewer 2 Report

This review manuscript is a thorough and comprehensive summary of the various approaches to modulate the blood-brain barrier (BBB) for improving drug delivery to the brain. To the reviewer’s knowledge, this is the first summary of its kind on this topic and a notable piece of work that will most certainly raise interest in the brain barriers field and be frequently cited. The manuscript is well-written, well-structured, to the point, and is an easy and enjoyable read.

Minor comments:

The overall tone of this review seems a bit too optimistic and one-sided. The reality is that despite the 50 years of research, we don’t have anything that allows safe opening of the BBB and/or that actually works. One could be critical and say it’s been 50 years of failure. Suggest adding some more literature that provides a counterbalance to the overly optimistic tone, especially in the FUS section. Yes, a lot of work has been done in this area but most of it unsuccessfully. In addition, some work shows that FUS leads to inflammation (e.g., Kovacs et al., PNAS, 2017 and others). Also, what one does not read about in the literature: from own experience, this reviewer has seen what rodent brain tissue looks like after FUS – red, 5mm-diameter hematoma-like spots, changes in brain “texture” – similar to what is observed after certain TBI.

Especially in the Future Perspectives (starting line 777), the authors should be a bit more balanced. Many of the modulators/approaches they listed are not in use any longer. Yes, FUS is currently being hyped up, but we have seen this with many other approaches over the last decades as well. The most promising brain drug delivery approach right now are probably delivery platforms targeting the TfR (see Ullman et al. Sci Transl Med, 2020, https://pubmed.ncbi.nlm.nih.gov/32461331/ and Kariolis et al., Sci Transl Med, 2020, https://pubmed.ncbi.nlm.nih.gov/32461332/)

The authors should include in their manuscript the publication by de Gooijer et al., Cell Rep Med, 2021 (https://pubmed.ncbi.nlm.nih.gov/33521698/). These authors discuss how an open BBB still does not allow drugs to enter due to the efflux transporters P-gp/ABCB1 and BCRP/ABCG2 (and there are maybe others such as Mrp4/ABCC4 that contribute to this phenomenon).

Line 29: drugs – should be drug’s

Suggest taking out some of the adverbs, they don’t add much. For example:

Line 48: highly restricted

Line 50: exceptionally high

Line 184: meaningful results – take out adjective. “meaningful” is subjective and has a different meaning for each reader.

Fig 1: OAT3 is shown on the luminal membrane but should be on the abluminal membrane. In addition, the figure legend mentions OATP3 – this is not the same as OAT3. Also, there is only one BCRP (not a BCRP1) – see also line 414.

The figures representing the BBB (e.g., Fig 1) could be drawn a bit more realistic. For example, brain capillaries are 5-7 microns in diameter, the endothelium itself is not more than 1 micron thick. Assuming a 7-micron capillary diameter, this would leave a 5-micron lumen. The figures shown have an endothelium that is almost as thick as the lumen diameter.

Lines 90-91 – this is too optimistic. The reality is that GBM median survival is 18-23 months (with treatment). This does certainly not speak to the efficacy of these drugs.

Line 114: FUS is not a therapy, it is an approach to open the BBB.

Line 149: by Hynynen et al…

Line 241: take out “of the

Line 463: suggest using Akt instead of PKB (Akt seems to be more common)

Suggest including the paper by Breitkreuz-Korff et al., 2021, J Control Release (https://pubmed.ncbi.nlm.nih.gov/34384796/). These authors propose a new Cldn-5 interaction inhibitor.

Fig 6: OAT3 is listed as efflux transporter, this is incorrect. For a comprehensive review on BBB efflux transporters see Hartz and Bauer, Curr Pharm Biotechnol, 2011 (https://pubmed.ncbi.nlm.nih.gov/21118088/)

Author Response

Reviewer 2:

Comments and Suggestions for Authors

This review manuscript is a thorough and comprehensive summary of the various approaches to modulate the blood-brain barrier (BBB) for improving drug delivery to the brain. To the reviewer’s knowledge, this is the first summary of its kind on this topic and a notable piece of work that will most certainly raise interest in the brain barriers field and be frequently cited. The manuscript is well-written, well-structured, to the point, and is an easy and enjoyable read.

Minor comments:

The overall tone of this review seems a bit too optimistic and one-sided. The reality is that despite the 50 years of research, we don’t have anything that allows safe opening of the BBB and/or that actually works. One could be critical and say it’s been 50 years of failure. Suggest adding some more literature that provides a counterbalance to the overly optimistic tone, especially in the FUS section. Yes, a lot of work has been done in this area but most of it unsuccessfully. In addition, some work shows that FUS leads to inflammation (e.g., Kovacs et al., PNAS, 2017 and others). Also, what one does not read about in the literature: from own experience, this reviewer has seen what rodent brain tissue looks like after FUS – red, 5mm-diameter hematoma-like spots, changes in brain “texture” – similar to what is observed after certain TBI.

Many thanks to Reviewer 2 for providing their own personal experience and expertise. In this light, we reviewed the literature reporting sterile inflammation and included a more balanced discussion within our future perspectives section (Pg 33 – BBB modulation & Sterile Inflammation) to provide insight into this reported phenomenon. While inflammation was reported primarily in FUS studies, one study also reported inflammation in mannitol-based BBB disruption. As noted by the reviewer, we suggest this phenomenon demands careful and diligent monitoring within future clinical trials.

In addition to this section we also lowered the overly-optimistic tone when discussing the completed FUS phase I clinical trials having considered the new literature published in this field.

Alterations:

  • Changed ‘There were signs of early microgliosis and astrocytosis at the target site following MRgFUS, however this resided 48 hours after administration.’ to ‘There were signs of early microgliosis and astrocytosis at the target site following MRgFUS that resided 48 hours after administration.’ line 198.
  • Removed ‘however, only for a short timeframe. The exact reason for this loss of expression within the liver was not identified, however previous reports suggest this was due to an immune-mediated effect’ from line 200
  • Removed ‘(17% increase in population survival time relative to trastuzumab only administration). A longer study however may demonstrate additional benefit.’ from line 208
  • Removed ‘Although no therapeutic findings were observed in this study, there was also no significant clinical or radiographic adverse events.’ from line 212
  • Changed ‘highlighting the safety of MRgFUS BBB modulation in humans.’ to ‘suggesting short-term safety within humans.’ Line 220.
  • Removed ‘great’ from line 221
  • Added ‘reportedly’ to line 229

We thank reviewer 2 for bringing this to our attention. It has significantly strengthened this review in our opinion.

Especially in the Future Perspectives (starting line 777), the authors should be a bit more balanced. Many of the modulators/approaches they listed are not in use any longer.

Although RMP-7 was a significant modulator two decades ago, it hasn’t shown any sign of rejuvenated research in recent years. We agree with Reviewer 2 that it has no place in the Future perspectives section of review. (Lines 838-840 of first review draft deleted).

Lines 838-840:

‘The only reported peptide to enter clinical trials for BBBD is RMP-7. Despite its safety as a BBB modulator, RMP-7 failed to meet the primary endpoint in in phase II trials although encouraging trends in the survival analysis were observed.’

es, FUS is currently being hyped up, but we have seen this with many other approaches over the last decades as well. The most promising brain drug delivery approach right now are probably delivery platforms targeting the TfR (see Ullman et al. Sci Transl Med, 2020, https://pubmed.ncbi.nlm.nih.gov/32461331/ and Kariolis et al., Sci Transl Med, 2020, https://pubmed.ncbi.nlm.nih.gov/32461332/)

Included both papers within a sub section of ‘Antibodies’ (lines 689-695). We also referenced a Research Highlight from Nature Reviews Drug Discovery that holds this BBB modulation approach in high regards, in agreement with Reviewer 2’s views.

The authors should include in their manuscript the publication by de Gooijer et al., Cell Rep Med, 2021 (https://pubmed.ncbi.nlm.nih.gov/33521698/). These authors discuss how an open BBB still does not allow drugs to enter due to the efflux transporters P-gp/ABCB1 and BCRP/ABCG2 (and there are maybe others such as Mrp4/ABCC4 that contribute to this phenomenon).

Incorporated this key paper into review (Pg 33 A Combined Approach Section)

Line 29: drugs – should be drug’s

Corrected as requested

Suggest taking out some of the adverbs, they don’t add much. For example:

Line 48: highly restricted

Corrected as requested

Line 50: exceptionally high

Corrected as requested

Line 184: meaningful results – take out adjective. “meaningful” is subjective and has a different meaning for each reader.

Changed to noteworthy

Fig 1: OAT3 is shown on the luminal membrane but should be on the abluminal membrane. In addition, the figure legend mentions OATP3 – this is not the same as OAT3. Also, there is only one BCRP (not a BCRP1) – see also line 414.

Corrected OATP3 errors through out various diagrams & legends, Moved OAT3 to abluminal membrane in Fig 1, Changed all instances of BCRP1 to BCRP.

The figures representing the BBB (e.g., Fig 1) could be drawn a bit more realistic. For example, brain capillaries are 5-7 microns in diameter, the endothelium itself is not more than 1 micron thick. Assuming a 7-micron capillary diameter, this would leave a 5-micron lumen. The figures shown have an endothelium that is almost as thick as the lumen diameter.

We narrowed the endothelial cells in Figure 1. Thinning the endothelial cell within the other diagrams displaying the NVU (Figs 4-6) is difficult due to the complex pathways elucidated in these diagrams, requiring a wide cell to include all information.

Lines 90-91 – this is too optimistic. The reality is that GBM median survival is 18-23 months (with treatment). This does certainly not speak to the efficacy of these drugs.

Removed Methotrexate paper as it didn’t strictly compare to current standard GBM treatment. We kept Trastuzumab as it did compare to the current standard, yet reduced the optimistic tone. See lines 93-95

Line 114: FUS is not a therapy, it is an approach to open the BBB.

Corrected as requested

Line 149: by Hynynen et al…

Corrected as requested

Line 241: take out “of the

Corrected as requested

Line 463: suggest using Akt instead of PKB (Akt seems to be more common)

Changed as requested

Suggest including the paper by Breitkreuz-Korff et al., 2021, J Control Release (https://pubmed.ncbi.nlm.nih.gov/34384796/). These authors propose a new Cldn-5 interaction inhibitor.

Excellent suggestion, happy to include! see M01 section (line 476)

Fig 6: OAT3 is listed as efflux transporter, this is incorrect. For a comprehensive review on BBB efflux transporters see Hartz and Bauer, Curr Pharm Biotechnol, 2011 (https://pubmed.ncbi.nlm.nih.gov/21118088/)

Noted and changed to ‘transporter’

Reviewer 3 Report

The manuscript ‘Modulating the Blood-Brain Barrier: A Comprehensive Review’ by Whelan et al. has been reviewed.

The review describes the structures, the mechanisms of action, the advantages and the drawbacks of the Blood-Brain Barrier modulators discovered and developed in the last fifty years.

Even if the argument is of interest, the authors list a large number of studies and reviews, but only superficially discuss the studies and data without providing much insight. To make the study interesting to the readers authors should include a critical analysis.

The manuscript presents several imprecisions, please find below just few examples:

-Typos: Line 29 “A drugs BBB permeability”, line 123, Line 381, Line 504 “high grade glioma”, page 32 “monoculture BBB” (is repeated three times), Line 771 “Van Itallie”, Line 837 “endpoint in in phase II trials”, page 38 “Blood brain barrier” and Parkinsons disease dementia”

- To make the paper clearer for the readers the figures and the tables should be placed after their citation in the text

- The quality of the figures should be improved, in particular Fig 1 (the names of channels and proteins are not readable).

-The literature style should be carefully uniformed, in addition many references are incompleted.

In my opinion, the reported work should be rejected as it stands, in fact is fundamental to add a critical analysis of the literature. The manuscript should be accepted after major revisions noted.

Author Response

Reviewer 3:

Comments and Suggestions for Authors

The manuscript ‘Modulating the Blood-Brain Barrier: A Comprehensive Review’ by Whelan et al. has been reviewed.

The review describes the structures, the mechanisms of action, the advantages and the drawbacks of the Blood-Brain Barrier modulators discovered and developed in the last fifty years.

Even if the argument is of interest, the authors list a large number of studies and reviews, but only superficially discuss the studies and data without providing much insight. To make the study interesting to the readers authors should include a critical analysis.

We thank reviewer 3 for including an honest and constructive review.

The combined feedback from all reviewers has provides us with a wide range of constructive information and comments. Combining the suggestions provided by all reviewers in this regard drove us to deliver two new sections within the future perspectives which provide additional insight into the area:

  • A Combined Approach (pg 33)
    • To furtherly promote the importance of ABC transporters in BBB modulation, we incorporated this key paper by de Gooijer et al (https://pubmed.ncbi.nlm.nih.gov/33521698/) as suggested by another reviewer that highlights the significant influence of P-gp in drug delivery to the CNS even when the BBB is opened paracellularly. By incorporating this paper within the review, we discuss how efflux transporter inhibition in combination with paracellular modulation is an important consideration to optimise the efficacy of BBB modulation hence drug delivery to the brain.

  • BBB Modulation & Sterile Inflammation (pg 33)
    • We reviewed the literature reporting sterile inflammation and included a heavy discussion within our future perspectives section (Pg 33 – BBB modulation & Sterile Inflammation) to provide insight into this reported phenomenon. While inflammation was reported primarily in FUS studies, one study reported inflammation in mannitol-based BBB disruption. We suggest this phenomenon demands careful and diligent monitoring within future clinical trials.

  • In addition to these two new sections, we also discuss the inherently facile degradation of peptides and why the development of a recently published small molecule gains advantage over cld-5 targeting peptides. (lines 477-487)

The manuscript presents several imprecisions, please find below just few examples:

 -Typos: Line 29 “A drugs BBB permeability” Corrected, line 123 Unable to see imprecision, Line 381 Corrected, Line 504 Corrected “high grade glioma”, page 32 “monoculture BBB” (is repeated three times) Corrected Line 771 “Van Itallie” Corrected, Line 837 “endpoint in in phase II trials” Section containing this sentence removed in line with feedback from another reviewer, page 38 “Blood brain barrier” Corrected every instance of ‘Blood brain barrier’ within the the clinical trial tables 5 + 6 to ‘Blood-Brain Barrier’ and Parkinsons disease dementia” Corrected.

 - To make the paper clearer for the readers the figures and the tables should be placed after their citation in the text We moved Table 3 from 1st draft (Now table 4 in new draft) further down the review so it is past all the modulators discussed to provide a clearer presentation and flow.

- The quality of the figures should be improved, in particular Fig 1 (the names of channels and proteins are not readable). Changed picture format to PNG for all diagrams and improved image resolution.

-The literature style should be carefully uniformed, in addition many references are incompleted. Referencing addressed and corrected

In my opinion, the reported work should be rejected as it stands, in fact is fundamental to add a critical analysis of the literature. The manuscript should be accepted after major revisions noted.

Many thanks to reviewer 3 for their valuable comments and suggestions. We feel we have now addressed each one and agree that it has significantly improved the impact of the review.

Reviewer 4 Report

Whelan et al. review entitled Modulating the Blood-Brain Barrier: A Comprehensive Review summarizes and compares various classes of Blood-Brain Barrier (BBB) modulators developed over the past five decades covering advancements, advantages and disadvantages along with giving some insight into the future of these modulators.

The authors highlighted the importance of BBB modulation and if successful it would enable larger and more polar molecules to reach the central nervous system which would enable more effective drugs to enter clinical trials.

The review includes an extensive overview of BBB modulators during the past 50 years highlighting the high failure rate of CNS drug candidates in drug development calling for further investigations into additional targets and pathways that may offer gains in selectivity.

The manuscript is well written and carefully produced with a clear language and informative tables to facilitate for the reader to easily grasp the different studies and their findings.

Please review the references in the manuscript, there are several references which do not have a reference number attached to the authors, e.g. row 269, Joshi et al., rows 346 and 352, Erdlenbruch et al, rows 366 and 371, Hülper et al, and others.

Author Response

Reviewer 4:

Comments and Suggestions for Authors

Whelan et al. review entitled Modulating the Blood-Brain Barrier: A Comprehensive Review summarizes and compares various classes of Blood-Brain Barrier (BBB) modulators developed over the past five decades covering advancements, advantages and disadvantages along with giving some insight into the future of these modulators.

The authors highlighted the importance of BBB modulation and if successful it would enable larger and more polar molecules to reach the central nervous system which would enable more effective drugs to enter clinical trials.

The review includes an extensive overview of BBB modulators during the past 50 years highlighting the high failure rate of CNS drug candidates in drug development calling for further investigations into additional targets and pathways that may offer gains in selectivity.

The manuscript is well written and carefully produced with a clear language and informative tables to facilitate for the reader to easily grasp the different studies and their findings.

Please review the references in the manuscript, there are several references which do not have a reference number attached to the authors, e.g. row 269, Joshi et al., rows 346 and 352, Erdlenbruch et al, rows 366 and 371, Hülper et al, and others.

Many thanks to the reviewer for their helpful suggestions and comments. We have addressed and corrected all issues concerning referencing throughout the review.

Round 2

Reviewer 3 Report

The manuscript should be accepted as it stands.